# Microparticle traction force microscopy reveals subcellular force exertion patterns in immune cell–target interactions

Daan Vorselen[1,2], Yifan Wang[3], Miguel M. de Jesus [4], Pavak K. Shah[5], Matthew J. Footer[1,2], Morgan Huse[4], Wei Cai [3] & Julie A. Theriot [1,2]*

Force exertion is an integral part of cellular behavior. Traction force microscopy (TFM) has been instrumental for studying such forces, providing spatial force measurements at sub-cellular resolution. However, the applications of classical TFM are restricted by the typical planar geometry. Here, we develop a particle-based force sensing strategy for studying cellular interactions. We establish a straightforward batch approach for synthesizing uniform, deformable and tuneable hydrogel particles, which can also be easily derivatized. The 3D shape of such particles can be resolved with superresolution (<50 nm) accuracy using conventional confocal microscopy. We introduce a reference-free computational method allowing inference of traction forces with high sensitivity directly from the particle shape. We illustrate the potential of this approach by revealing subcellular force patterns throughout phagocytic engulfment and force dynamics in the cytotoxic T-cell immunological synapse. This strategy can readily be adapted for studying cellular forces in a wide range of applications.

[1] Department of Biochemistry, Stanford University, Stanford, CA 94305, USA. [2] Department of Biology and Howard Hughes Medical Institute, University of Washington, Seattle, WA 98105, USA. [3] Department of Mechanical Engineering, Stanford University, Stanford, CA 94305, USA. [4] Immunology Program, Memorial Sloan Kettering Cancer Center, New York, NY 10065, USA. [5] Developmental Biology Program, Sloan Kettering Institute, New York, NY 10065, USA. *email: jtheriot@uw.edu

Cells continuously exert forces on their environment, allowing them to probe and respond to their mechanical surroundings[1], while also engaging in interactions with neighboring cells[2–4]. Traction force microscopy (TFM) provides the ability to study such forces in detail[5,6]. However, classical TFM is largely limited to in vitro applications[7], and a range of cellular processes warrant different (i.e., non-planar) geometries, such as cell–cell interactions like phagocytosis or the formation of the immunological synapse, in which mechanical forces have a critical role[8,9]. Moreover, with a few notable exceptions[10–13], classical TFM focuses on the study of shear forces, often neglecting forces normal to the surface.

Force sensing using spherical sensors is a new area of exploration and first became possible within tissue, and in vivo, by the introduction of oil droplet-based technologies[14–16]. Recently, deformable hydrogel microparticles (MPs) were also used as force reporters[17]. The use of microgels for MP based traction force microscopy (MP-TFM) shows promise for broad applicability because of the mechanical tuneability over orders of magnitude and the potential to measure both normal and shear stresses. Moreover, these elastic microparticles likely more closely mimic cellular mechanical properties than fluid droplets, making them more appropriate model systems for studying cell–cell interactions. However, current technologies have lacked the resolution to identify contributions from individual subcellular force transmitting structures. Moreover, synthesis of suitable hydrogel MPs for such applications can be rather complex, often requiring specialized expertise in microfluidics[17,18]. Particle properties, especially their size and mechanical properties, must also be optimized for accurate traction force measurements[18].

Here, we describe a particle-based force sensing strategy specifically designed for studying ligand-dependent cellular interactions at high spatial resolution (Supplementary Fig. 1). We established a batch production approach for synthesizing cell-sized deformable, and tuneable, hydrogel particles that is simple and reproducible. The resulting particles are homogeneous throughout their volume and from particle to particle. The incorporation of carboxylic acid groups allows easy and efficient derivatisation. Using confocal microscopy, we can reconstruct the particle boundary with superresolution accuracy, allowing us to visualize cell-induced local deformations with <50 nm precision. Finally, we solve the inverse problem of inferring the displacement field and traction forces from the measured particle shape by iteratively minimizing a cost function consisting of contributions from the residual tractions on the traction-free region and the elastic energy, using a fast spherical harmonics-based method[19]. We illustrate the potential of this MP-TFM method by revealing subcellular details of the mechanical interaction of macrophages with their targets during phagocytosis, as well as the dynamics of force exertion in the T-cell immunological synapse.

## Results

**Batch production of deformable hydrogel microparticles.** We used membrane emulsification with Shirasu Porous Glass (SPG) membranes with uniform pore size to produce monodisperse spherical hydrogel microparticles[20,21]. Acrylamide mixtures containing monomeric acrylamide (AAm), acrylic acid (AAc) and the crosslinker $N,N'$-methylenebisacrylamide (BIS) were extruded through a hydrophobic SPG membrane under nitrogen pressure, generating droplets in an oil phase (Fig. 1a). After the entire aqueous mixture was dispersed, acrylamide polymerization was initiated within the droplets by addition of the free radical initiator Azobisisobutyronitrile (AIBN). The resulting deformable poly-AAm-co-AAc microparticles (DAAM-particles) were

collected and resuspended in PBS. As shown previously[22,23], this strategy yields highly monodisperse particles (Fig. 1b). The particle size could be tuned between 4–15 μm (each with coefficient of variation (CV) $\lesssim 0.1$) using SPG membranes with various pore sizes, revealing the expected linear relation between pore and particle size[23] (Fig. 1c). Furthermore, since membrane emulsification is a bulk method, we were able to synthesize an estimated $10^{10}$–$10^{11}$ particles per batch (Supplementary Fig. 2).

To further tune important particle properties, such as their rigidity, we changed the composition of the acrylamide mixture. In particular, the total mass concentration of acrylic compounds ($C_T = C_{AAm} + C_{AAc} + C_{BIS}$) was kept constant, while the cross-linker concentration $C_c = m_{BIS}/(m_{AAm} + m_{AAc} + m_{BIS})$ was varied between 0.3 and 2.3% (Supplementary Fig. 2). As expected, $C_c$ affects the hydrogel swelling properties, leading to an inverse relation between $C_c$ and particle size (Fig. 1d).

Particle functionalization is critical for triggering specific cellular behavior and required for visualization of the particles and their deformation in microscopy applications. The incorporation of acrylic acid in the gel mixture allowed straightforward and efficient covalent coupling of primary amine groups to the DAAM-particles using carbodiimide crosslinking chemistry[24]. DAAM-particles were functionalized with both fluorescently conjugated bovine serum albumin (FITC-BSA) and small fluorescent molecules (TRITC-Cadaverine (Cad)) (Fig. 1e). Confocal imaging revealed highly homogeneous coupling of both protein and reactive dye to the microporous DAAM-particles.

The ideal force-reporting microparticle has a homogeneous and isotropic network structure, which underlies equally homogeneous elastic properties. However, gel swelling or non-uniform polymerization could potentially lead to a radially inhomogeneous network structure. Derivatization of the particles could also affect particle mechanical properties and lead to further inhomogeneities. Therefore, we evaluated the radial fluorescent intensity profile ($n > 100$ particles) of both FITC-BSA and TRITC-Cad conjugated to DAAM-particles (Fig. 1f, Supplementary Figs. 2 and 3). This revealed no radial deviations from a homogeneous solid sphere, even for the softest particles ($C_c$ 0.32%), which have swollen significantly (~5-fold) from the emulsion droplet size. Combined with the homogeneous appearance of individual particles, and osmotic shrinking experiments (Supplementary Fig. 3), this strongly suggests that these particles have uniform and isotropic polymer networks.

In addition to intraparticle homogeneity, strong interparticle uniformity is critical for the application of microparticles as force sensors. To evaluate particle-to-particle variation, we measured quantitative 3D refractive index (RI) maps of individual DAAM-particles (Fig. 1g). These measurements revealed RI values of $1.3349 \pm 0.0001$ (standard deviation (s.d.), $n = 8$ particles) to $1.3381 \pm 0.0001$ (s.d., $n = 29$ particles) for particles with $C_c$ 0.32% and $C_c$ 2.3%, respectively (Fig. 1g). The coefficient of variation (CV: calculated as the standard deviation divided by the relative RI difference between particle and medium) within each condition were markedly low at 0.03–0.1. The similarity of the standard deviations between samples suggests that the observed variation may be dominated by the measurement error, and the particle-to-particle variation in polymer network density is potentially even lower.

The close match of RI between these particles (1.3349–1.3381) and the medium ($RI_{PBS} = 1.334$) also represents a significant benefit over glass ($RI_{GLASS} = 1.45$) or polystyrene (PS, $RI_{PS} = 1.59$) beads for many microscopy applications, including their use as target models for immune cell–target interactions. Optical distortions, i.e., the reflection and refraction of light, caused by a particle depends on $RI_{medium}$–$RI_{MP}$. For DAAM-particles this value is 0.001–0.005, whereas for glass or polystyrene these values

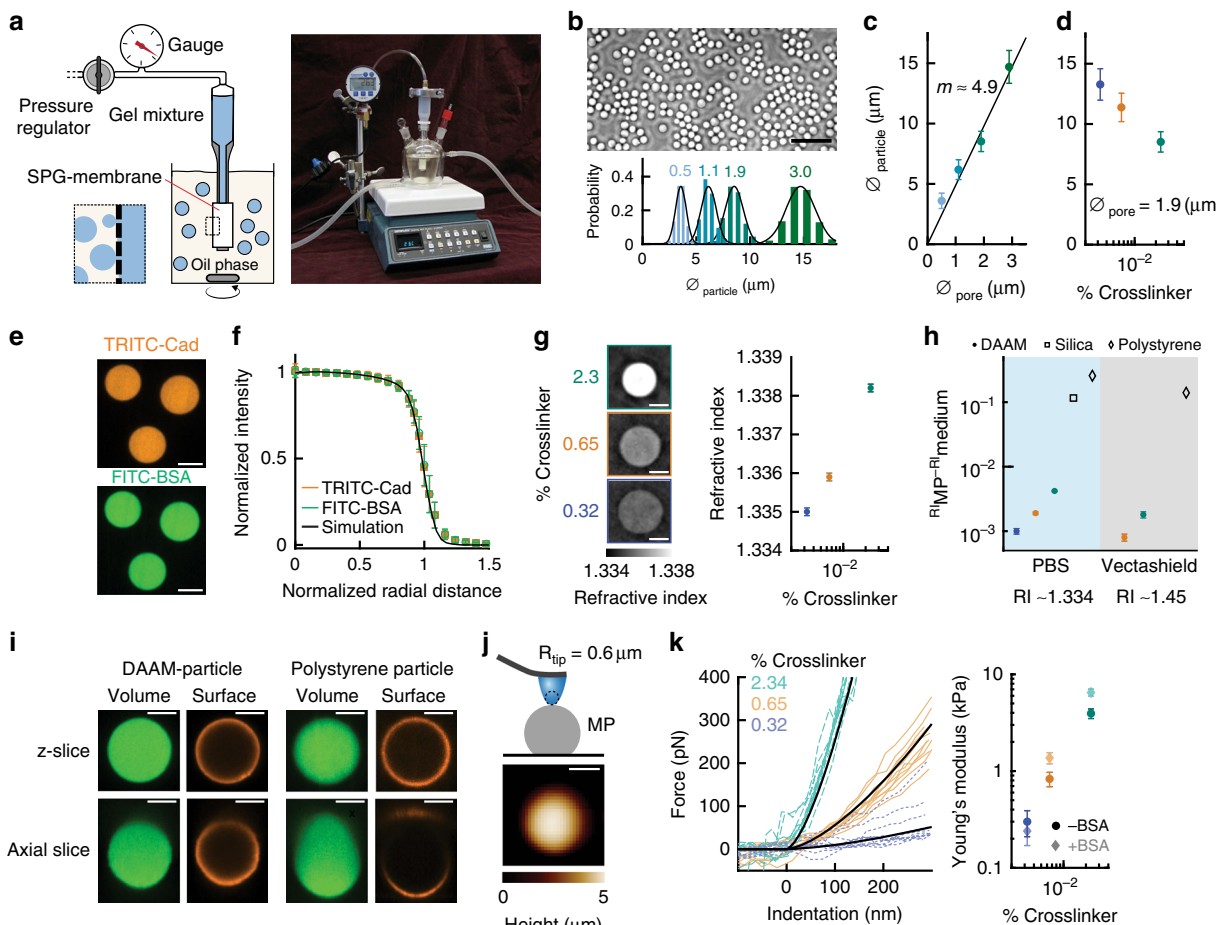

**Fig. 1 Characterization of batch-produced microparticles with cell-like mechanical properties. a** Schematic diagram and photograph of the components for membrane emulsification-based microparticle synthesis. **b** (Top) Phase image of DAAM-particles obtained with 1.9 μm SPG membranes (crosslinker concentration $C_c$ 2.3%). Scale bar is 50 μm. (Bottom) DAAM-particle diameter distributions from 0.5–3.0 μm SPG (colored numbers) ($n > 500$ particles per condition). Black lines represent Gaussian fits. **c** Particle diameter dependence on SPG pore diameter. Black line is a linear fit through the origin. **d** Particle size dependence on $C_c$ ($n > 400$ particles per condition). **e** Confocal z-slices of double functionalized DAAM-particles. **f** Radial intensity profiles for $C_c$ 2.3%. >100 profiles were each normalized to intensity and radius, and then averaged. Simulated data represents a homogeneous sphere convolved with an experimental PSF. **g** DAAM-particle refractive index ($RI_{MP}$) in PBS (RI 1.334) (Left) 3D quantitative RI tomogram slices; (Right) Quantification ($n = 7$, 14 and 28 particles for $C_c$ 0.32%, 0.65% and 2.3%, respectively; **h** RI difference between particle and medium ($RI_{medium}$) in PBS and Vectashield (VS, RI 1.45) ($n = 9$ and 12 for $C_c$ 0.65% and 2.3%, respectively). Filled and open markers indicate measured and manufacturer-specified values, respectively. Silica and $C_c$ 0.32% DAAM-particles in VS are missing, as $RI_{MP} \approx RI_{medium}$. The latter were practically impossible to visualize. **i** Confocal slices through a DAAM-particle ($C_c$ 0.32%) and a 10 μm polystyrene particle in PBS. DAAM-particles were functionalized with Alexa Fluor-488-Cadaverine. Surface of both particle types was functionalized with BSA and then immunostained. **j** (Top) Schematic diagram of DAAM-particle mechanical characterization by AFM. (Bottom) Typical AFM image of a DAAM-particle ($C_c$ 2.3%). **k** (Left) Force-indentation curves (FDCs) on DAAM-particles functionalized with BSA ($n = 11$ particles per $C_c$). 2–5 FDCs were made per DAAM-particle and for clarity only the FDC closest to that particles' average behavior is visualized. (Right) DAAM-particles Young's moduli with and without BSA ($n = 11$, 14 and 12 for $C_c$ 0.32%, 0.65% and 2.3%, respectively). All error bars indicate standard deviations. All white scale bars are 5 μm. Source data are provided as a Source Data file.

are typically 100-fold higher (Fig. 1h). We compared confocal images of DAAM and PS particles with volumetric and surface labels (Fig. 1i). Z-slices of both particles revealed apparently equivalent signals. Axial slices of the volume of PS particles, however, showed a distorted shape (the upper hemisphere appears axially elongated by $\sim RI_{medium}/RI_{MP}$) and an inhomogeneous signal. Furthermore, the fluorescent signal from the upper hemisphere's surface is reflected off the PS-particle surface and is largely not captured on the camera. Such lensing effects are barely noticeable for DAAM-particles. Finally, due to the porous nature of DAAM-particles, the advantageous optical properties are rather independent of the medium, as evidenced from repeating these measurements in Vectashield with RI 1.45 (Fig. 1h, Supplementary Fig. 5).

To allow quantitative force sensing with deformable DAAM-particles, we characterized their mechanical properties by atomic force microscopy (AFM). We used pyramidal tips with a large end radius (~600 nm), which allow imaging of spherical particles, and indentations in the Hertzian regime (Fig. 1j). Imaging of single particles before indentation allows estimation of the particle shape (Supplementary Fig. 6) and hence precise localization of the centroid of the particle for subsequent indentations (Fig. 1j). Nano-indentations at the particle's centroids showed typical Hertzian force responses and revealed cell-like rigidities, as quantified by the Young's moduli ($E_y$) between 0.3 and 3.9 kPa, for $C_c$ 0.32–2.3% (Fig. 1k). For the more rigid particles, we found up to 1.5-fold higher moduli after conjugation of the particles with BSA, which is likely due to

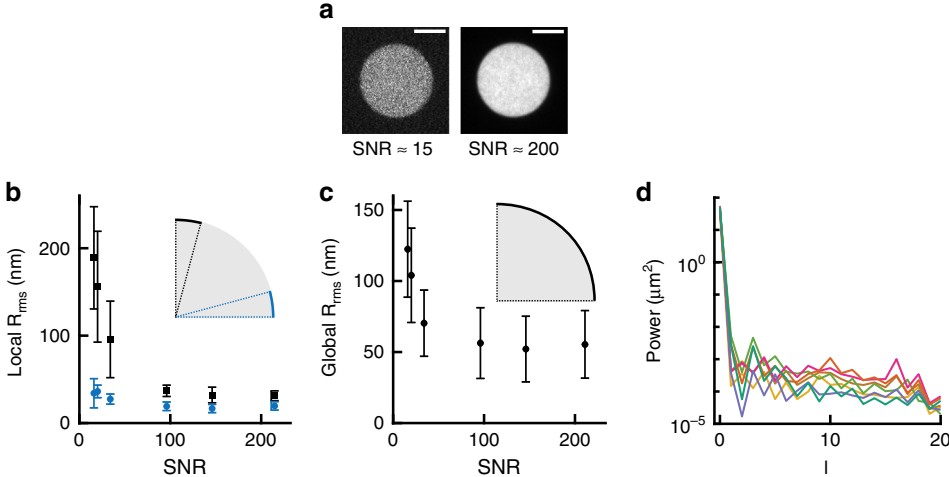

**Fig. 2 Superresolved edge localization and reference shape measurements of undeformed DAAM-particles. a** Confocal images (z-slices, 50 ms exposure per slice) of DAAM-particles (soft: $C_c$ 0.32%) in Vectashield (VS) taken with different laser power leading to various signal-to-noise ratios (SNR). SNR was calculated as $(\mu_{MP} - \mu_{bg})/s_{bg}$, where $\mu_{MP}$, $\mu_{bg}$ are the mean fluorescent intensity of the particles and background, respectively, and $s$ indicates the standard deviation of the background fluorescent intensity. **b** Quantification of edge localization precision as the root-mean-squared surface roughness $R_{rms}$ over a 1 μm² surface area ($n = 7$ DAAM-particles, the same set of particles is used for each condition). Localization precision near the equatorial plane (latitude < 15°) and at the upper apex (latitude > 75°), are visualized in blue and black, respectively (see insets). Only the upper hemispheres of DAAM-particles were analyzed to exclude the effects of deformation due to adhesion to the glass substrate. **c** Surface roughness over the entire upper hemispheres ($n = 7$ DAAM-particles). **d** Spherical harmonic power spectrum indicating the power of the distance to the centroid over the particle surface per degree $l$ (summed over orders $m$) for 7 particles; colors indicate individual particles. Source data are provided as a Source Data file.

increased crosslinking of the hydrogel by BSA binding to multiple carboxyl groups. Strikingly, the Young's moduli are 5–10-fold lower than bulk hydrogels of similar composition (Supplementary Fig. 6). Such discrepancy may be caused by the differences in polymerization conditions (e.g., different initiator and temperature) or gel swelling, and illustrate the importance of direct mechanical characterization of MPs. Finally, we also measured the bulk modulus of the DAAM-particles, which, together with the Young's modulus measurements, yielded Poisson's ratio estimates around 0.43 and revealed that the particles are close to incompressible (Supplementary Fig. 3).

**Superresolved shape analysis of DAAM-particles**. To determine the accuracy with which the particle shape can be reconstructed and to generate a reference shape measurement of undeformed particles, we imaged adherent particles in Vectashield mounting medium (RI ~ 1.45). The 3D shape of individual particles was reconstructed by processing confocal imaging stacks, encompassing deconvolution, edge detection, and edge localization steps (Supplementary Fig. 7). We varied the laser power to obtain particle images with various signal-to-noise ratios (SNR) (Fig. 2a) and estimated our localization precision as the local surface roughness $R_{rms}$ observed over a small area (1 μm²). Due to the non-isotropic resolution in confocal microscopy, the local $R_{rms}$ depended on the position on the particle. Therefore, we separately quantified the $R_{rms}$ at the equator (latitude < 15°) and at the particle apex (latitude > 75°), which are dominated by the resolution in xy and z, respectively (Fig. 2b). Even at modest SNR (~15) the localization precision in z (~200 nm) and xy (~40 nm) exceeded the nominal resolution (full width at half maximum of the PSF) in the corresponding direction. The principle enabling us to superresolve the particle boundary is similar to single molecule localization techniques[25,26] (Supplementary Discussion). With increasing SNR (>100), the localization further improved (z: ~40 nm, xy: ~20 nm), also leading to a 2.5-fold reduction in the resolution anisotropy (Fig. 2b).

Aside from edge localization precision, the accuracy with which deformations can be measured (and forces recovered) depends on the particle shape in the absence of externally applied forces. Deviations from a perfect spherical shape can, for example, be caused by deformation of the emulsion droplets during gelation. In addition, apparent shape deviations can arise from imaging artefacts like drift during recording of confocal stacks. To measure such deviations, we quantified 1) the global $R_{rms}$, calculated over the entire upper particle hemisphere and 2) the sphericity, defined as the surface area of a sphere with equal volume to the particle divided by the observed particle surface area (1 for a perfect sphere). Global $R_{rms}$ measurements revealed small (50 ± 25 nm, s.d., $n = 7$ particles) overall surface roughness, corresponding to deviations of less than 1% of the particle radius (Fig. 2c). Sphericity calculations corroborated that DAAM-particles are very spherical ($\psi = 0.9988 \pm 0.0004$, s.d., $n = 7$ particles). Finally, we decomposed the DAAM-particle shape using spherical harmonics for spectral analysis (Supplementary Note 1), which revealed that deviations from a perfect sphere have no characteristic length scale (white noise) (Fig. 2d).

**Local target deformations induced during phagocytosis**. Cellular forces have an important role during target engulfment in phagocytosis. Spatial distributions of phagocytic forces have previously only been measured in the frustrated state on flat substrates[27], and our conjugatable and deformable DAAM-particles present an ideal tool for measuring them on particles with more physiological curvature. We exposed J774 murine macrophage-like cells to soft DAAM-particles (0.3 kPa, $C_c$ 0.32%) that were labeled with TRITC-Cad and functionalized with BSA and anti-BSA immunoglobulin G (IgG). J774s showed strongly ligand-dependent attachment to, and internalization of, antibody-coated DAAM-particles (Supplementary Fig. 8). Moreover, in agreement with previous findings[8,28], phagocytic engulfment was strongly rigidity-dependent, with uptake of 7 kPa ($C_c$ 2.3%) DAAM-particles being ~6-fold more efficient than uptake of 1 kPa ($C_c$ 0.23%) particles (Supplementary Fig. 8). After fixation

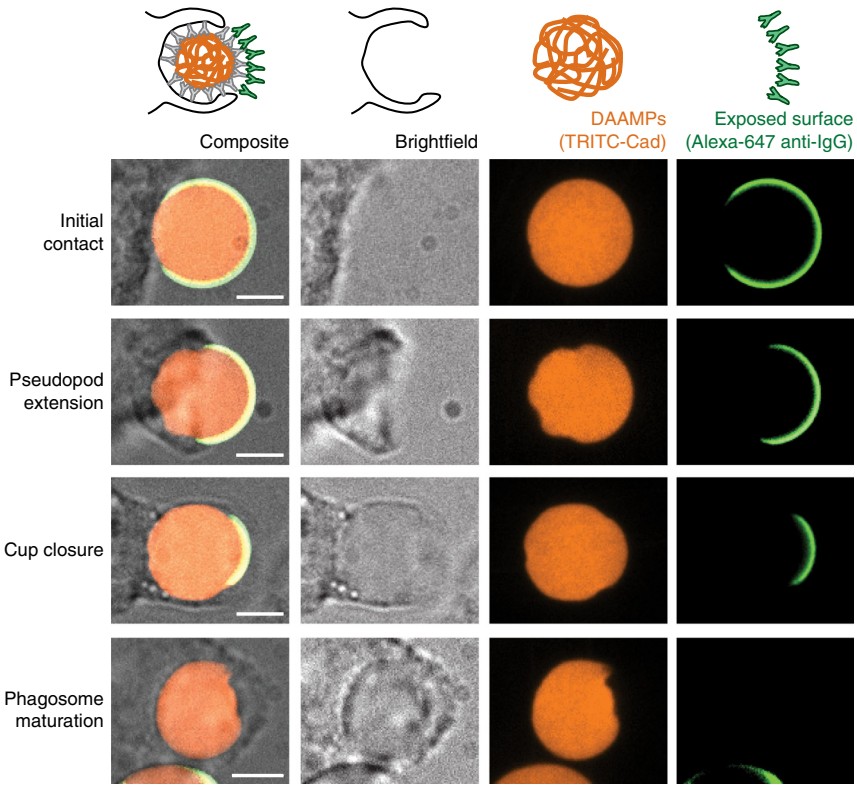

**Fig. 3 J774 macrophage-like cells deform DAAM-particles throughout IgG-mediated phagocytic engulfment.** Confocal images (z-slices) through the centroid of particles undergoing phagocytosis. DAAM-particles were functionalized with TRITC-Cadaverine (Cad), BSA and anti-BSA IgG. J774 murine cells were exposed to DAAM-particles for 5 min and subsequently fixed. After fixation, the freely exposed particle surface was immunostained with Alexa Fluor-647 secondary antibody. Images are representative examples from multiple (n > 5) independent experiments. All scale bars are 5 μm. Larger field of views encompassing the entire cells are provided in Supplementary Fig. 8.

of the cells, a secondary antibody was added, which is sterically unable to diffuse into the cell–target contact area[29]. Because the secondary antibody only binds the exposed DAAM-particle surface, it indicates the progression of the phagocytic process (Fig. 3, Supplementary Fig. 8). Confocal images of DAAM-particles undergoing phagocytosis revealed local cell-induced deformations at all stages of the process.

To comprehend the full extent of phagocytic target deformation by macrophage-like J774s, the 3D shape of representative particles in various stages of phagocytic engulfment was reconstructed (Supplementary Fig. 7; Fig. 4). In addition, the fluorescent intensity of the secondary antibody over the particle surface allowed determination of the contact area between the phagocyte and the DAAM-particle, and the location of the base of the phagocytic cup (Supplementary Fig. 8; Fig. 4). This analysis revealed complex and distinct patterns of deformation, suggesting an intricate series of mechanical interactions throughout phagocytosis.

**Reference-free estimation of normal and shear stresses.** To reveal the cellular forces exerted on phagocytic targets, we calculated the traction forces from the observed particle shapes. In classical TFM, the displacement field is measured directly, while in our method, the 3D shape of DAAM-particles is measured at high resolution instead. The surface displacement field is not uniquely determined by the surface shape, since multiple displacement fields can lead to the same shape. To derive traction forces, we thus solved the inverse problem of inferring the traction forces given the observed particle shapes and the traction-free regions from the fluorescent immunostaining

(Supplementary Fig. 7). This is accomplished by an iterative optimization procedure described below (Fig. 5a).

We started from nodes on the surface of the measured 3D shape (Supplementary Note 2). By assuming these mesh nodes being restricted on the measured shape surface, a trial displacement field $\mathbf{u}(\theta, \varphi)$ can be obtained, where the co-latitude and the longitude can be calculated by shifting the mesh nodes on the surface of the measured shape $(\theta, \varphi)_i = (\theta_0 + \Delta\theta, \varphi_0 + \Delta\varphi)_i$ (See Methods). Starting from $\mathbf{u}(\theta, \varphi)$ as the displacement boundary condition of the boundary value problem in small-strain linear elasticity regime, we can rapidly obtain the full solution, including the stress field $\boldsymbol{\sigma}(\theta, \varphi)$ everywhere in the particle, the traction force $\mathbf{T}(\theta, \varphi)$ on the surface, and the elastic energy $E_{el}$ using the spherical harmonics-based method[19]. This trial displacement solution guarantees the mesh nodes match the measured shape perfectly. The residual traction on the traction-free region is quantified by the function $R(\mathbf{T}; \partial\Omega_t)$ (see Methods). A cost function $f$ is then constructed for the trial solution $\mathbf{u}(\theta, \varphi)$:

$$f(\mathbf{u}) = E_{el} + \alpha R^2(\mathbf{T}; \partial\Omega_t) + \beta E_{pen}(\mathbf{T}) \tag{1}$$

where $\alpha$ is the weighing parameter for residual traction; $\beta E_{pen}$ is an anti-aliasing term, which is included to prevent over-estimation of high-frequency contributions in the force calculations (Supplementary Note 3). The cost function is iteratively minimized using the conjugate-gradient algorithm. The elastic energy is included in the cost function to penalize unphysical solutions that produce the same shape.

The minimization process requires evaluation of the cost function and its gradient for thousands of displacement fields $\mathbf{u}(\theta, \varphi)$. Therefore, each function and gradient value needs to be evaluated quickly. Using the spherical harmonics-based

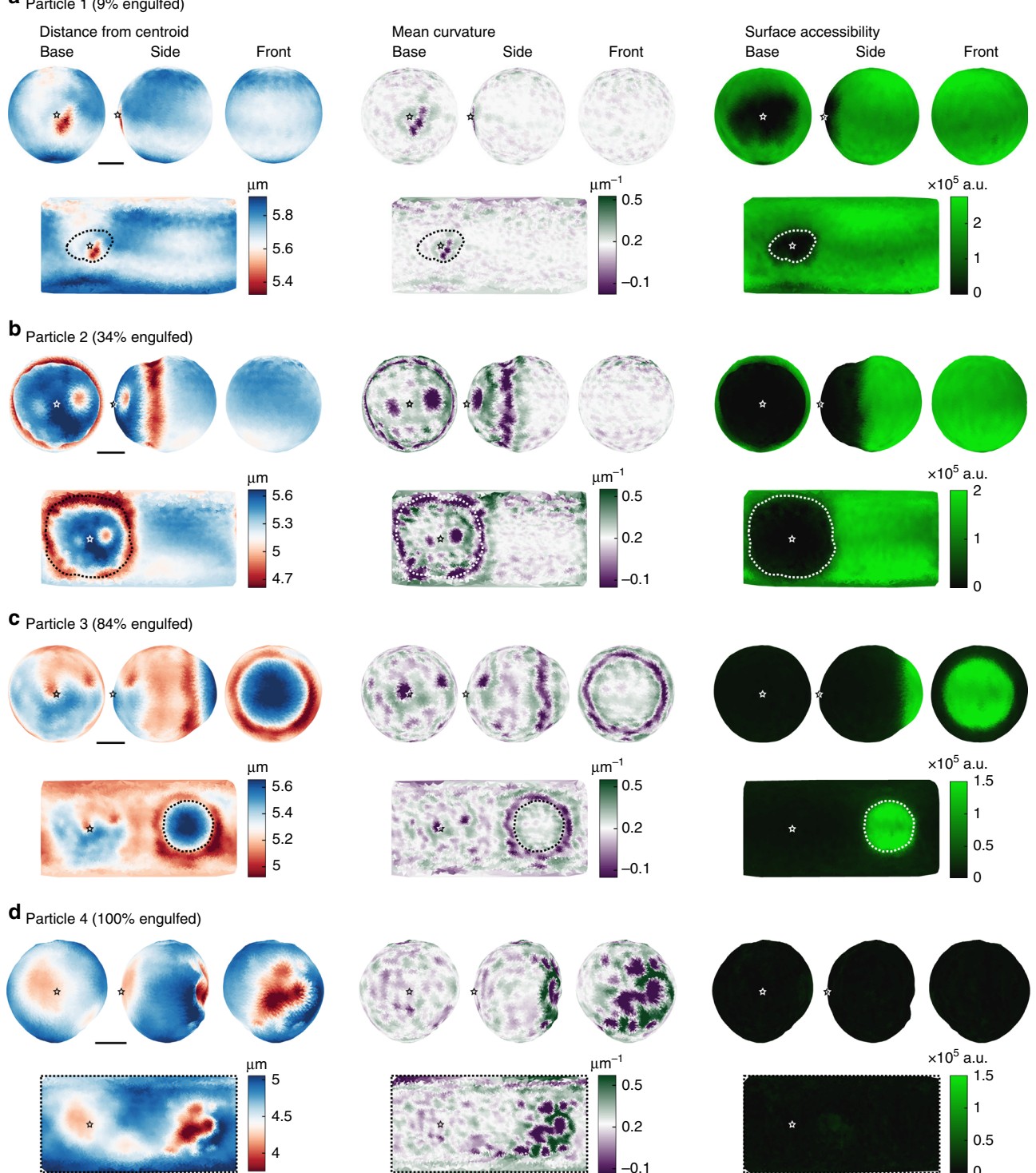

**Fig. 4 3D reconstruction of model targets deformed during phagocytic engulfment. a–d** Soft ($C_c$ 0.32%, $E_y$ 0.3 kPa) DAAM-particles imaged in various stages of phagocytic engulfment. Left, distance of the surface from the volumetric centroid. Middle, mean curvature of the particle surface. The color scale is centered around the median of the mean curvature (~1/$R$, with $R$ the particle radius). Right, fluorescence intensity (integrated along the radial direction) from immunostaining of the particle surface. Top rows, visualization of particle surfaces from three different viewpoints. Bottom rows, Equirectangular map projections (standard parallel taken at latitude 0) showing the full particle surface, although necessarily distorted (most strongly around the polar regions). Stars (white or black) mark the base of the phagocytic cups (as determined from the binarized secondary antibody signal), and dashed lines (white or black) mark the outlines of the phagocytic cups. Color scale legends are presented above each column. All scale bars are 3 μm.

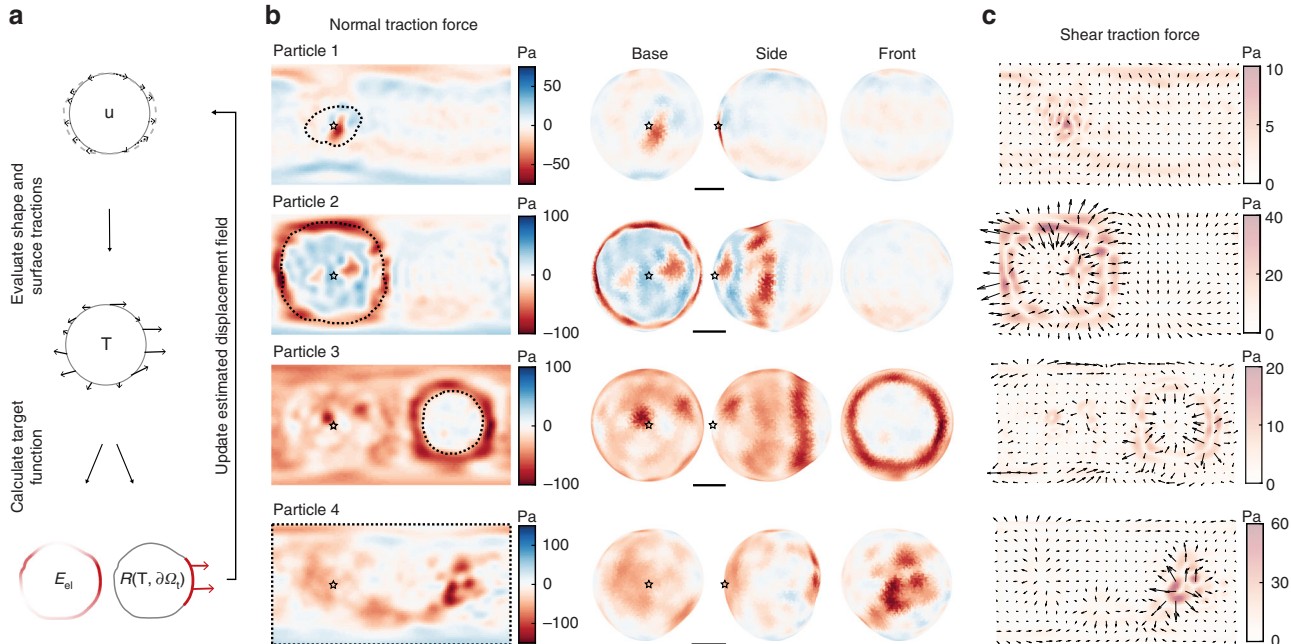

**Fig. 5 Direct computation of phagocytic traction forces from deformed particle shape. a** Schematic representation of the computational approach. The elements of the cost function that are minimized for calculation of cellular forces are highlighted in red. Dashed arrow indicate the deformation field, whereas thick solid arrows indicate forces. **b** Normal traction force reconstruction for soft DAAM-particles ($C_c$ 0.32%, $E_y$ 0.3 kPa) in various stages of engulfment. Left, equirectangular map projections (standard parallel taken at latitude 0) showing normal traction forces over the entire particle surface. The base of the phagocytic cup is marked with a black star. Right, visualization on the calculated particle shapes from 3 different viewpoints. Color scale with red color and negative values indicate inward compressive forces, whereas blue colors and positive values indicate outward tensile forces. **c** Shear traction forces visualized in equirectangular map projections (similar as in **b**). Arrows indicate the direction of the shear forces. Black stars mark the base of the phagocytic cups, and black dashed lines mark the outlines of the phagocytic cups. Color scale indicates the magnitude of the shear forces. Tractions were calculated with spherical harmonic coefficients up to $l_{max} = 20$, and evaluated on a $21 \times 41$ grid. For visualization, the resulting forces were (up) sampled with $61 \times 121$ pixel density using the same set of spherical harmonic coefficients. Top to bottom, same four particles are shown as in Fig. 4. Note that the calculated traction forces, especially the absolute magnitude, may be less accurate for particle 4 than for the other particles, due to the lack of a traction-free region (Supplementary Fig. 9). All scale bars are 3 µm.

method[19], we can solve the boundary value problem and obtain the cost function and its gradient within 0.2 s, so that the cost function can be minimized within 2 h. This would not be possible using general purpose elasticity solvers such as the finite element method (FEM).

To validate this computational strategy, we designed test cases based on in silico indentation of an elastic sphere by opposing forces. To accurately mimic experimental data, the test cases included normal and shear tractions, a traction free boundary, and noise comparable to our experimentally measured shapes (Supplementary Fig. 9). We first used these computational cases to find the optimal values for the weighing coefficients α and β (Supplementary Fig. 10), with which we could accurately recover both the direction and magnitude of the input traction forces. Notably, this held true for normal forces as well as for shear forces (Supplementary Fig. 9). Only small residual traction was present on the traction-free region, which was also the case for the experimental data ($R(\mathbf{T}; \partial\Omega_t) < 7$ Pa) (Fig. 5). Together, these results indicate accurate determination of traction forces using our computational methodology.

**High resolution study of phagocytic deformations and forces.** The method presented here allows studying cellular forces throughout phagocytosis and resulting target deformation in great detail, revealing contributions of individual subcellular force transmitting structures. Observations in the initial stage of phagocytosis (particle 1, 9% engulfed) provided evidence of the cell compressing its target ($C_c$ 0.32%, $E_y$ 0.3 kPa) in discrete spots

(Fig. 4a, Supplementary Movie 1). Computation of the mean curvature H over the particle surface clearly showed that the deformation induced by the cell consists of three distinct spots of ~1 µm in diameter. The compressive stresses causing this deformation were ~60 Pa (Fig. 5b). Approximately 9% of the DAAM-particle was covered by the cell and the compressive forces were located centrally in this area (Fig. 5b).

Significantly larger target deformations of up to 1 µm in the radial direction were observed during the stage of pseudopod extension (particle 2, 34% engulfed) (Fig. 4b, Supplementary Movie 2). The majority of compressive deformation by the cell was localized in a ring, which coincided closely with the extension of the pseudopod as estimated from the exposed surface labeling, while the indentation induced within the ring is markedly heterogeneous with clear dents and bumps ($R_{rms} = 120$ nm) (Fig. 4b, Supplementary Fig. 11). These deformations seem consistent with recently reported podosome-like structures during phagocytosis[30]. The heterogeneity appears to be caused by equally varied exertion of normal stresses from 20 to 100 Pa, totaling in 2.5 nN normal force (Fig. 5b). Shear stresses up to 50 Pa were also concentrated in this area, and were seemingly distributed in two separate, balanced rings: an outer ring with outward shear forces (0.5 nN total) and an inner ring with shear forces pointing to the cup base (0.5 nN total) (Fig. 5c). The ring-like compression was accompanied by a ~6% elongation in the direction normal to the ring. Whether such elongation is a result of orthogonal compression alone, or is, at least partly, caused by tension along the axis of elongation is difficult to assess from deformation analysis alone (Supplementary Fig. 9). The force

analysis, however, uses additional information such as the stress-free boundary and revealed, surprisingly, pulling forces throughout the base of the cup with tensile stresses of ~30 Pa (~1 nN total force). The entire engulfed surface exposed greater heterogeneity in curvature than the particle surface not in contact with the cell (Fig. 4b). In particular, two distinct pits were visible close to the base of the phagocytic cup (indentation depths: ~250 and ~700 nm). Such deformations were caused by localized compressive stresses up to 100 Pa, and were not expected since in this phase of phagocytosis the cytoskeletal actin is being cleared from the cup base[31].

Observations in the stage of cup closure (particle 3, 84% engulfed) indicated compression by the cell in a rather smooth ring, and an approximately 7% particle elongation normal to the ring (Fig. 4c, Supplementary Movie 3). Although deformations in the ring were less strong than observed in particle 2, the underlying normal stresses were mostly higher (60–100 Pa, 3.5 nN total force), which can be explained by compression of the target throughout the cell–target contact area (Fig. 5b). The smaller variation in force also resulted in a more homogeneous appearance of the ring ($R_{rms} = 60$ nm) than in particle 2 (Fig. 4c, Supplementary Fig. 11). Shear forces were significantly lower in this stage than in particle 2. Interestingly, the peak of the inward deformation and force was not localized at the cell boundary but trailing ~1 μm behind. Curvature and traction patterns were visible throughout the phagocytic cup, and seemed somewhat concentrated at the phagocytic cup base with 2 notable dents (~80 Pa, 0.2 nN normal force per dent). Such pits, which were also visible in particle 2, resemble the size and shape of podosomes[32,33], and could indicate a role for these mechanosensitive structures in phagocytosis, consistent with recent reports[30].

Finally, a fully internalized particle (particle 4), which lacked fluorescent immunostaining (Fig. 4d, Supplementary Movie 4), was subjected to some of the strongest indentations (~1 μm) by the cell. These strong normal stresses (~150 Pa) were also accompanied by large shear stresses up to 60 Pa (Fig. 5c), although due to the lack of a known stress-free region for this particle, our estimate of the magnitude of the forces may be less accurate for this particle (Supplementary Fig. 9). The deformations consisted of multiple (at least 8) dents, ranging between 1–2 μm in diameter. The brightfield image (Fig. 3) was used to estimate the location of the cup base, revealing that these compressive deformations arise from the site of phagocytic cup closure. Perhaps this indicates that actin that accumulates during cup closure[34] contributes to the repositioning of the maturing phagosome towards the center of the cell.

**Force dynamics in the cytotoxic T-cell immunological synapse.** Next, we applied MP-TFM to study the dynamics of cytotoxic T lymphocyte (CTL) force exertion at the immunological synapse (IS). Such forces are known to correlate with CTL cytotoxicity[9]. DAAM-particles ($C_c$ 0.32%, $E_y$ 0.3 kPa) were functionalized with intercellular adhesion molecule 1 (ICAM-1) and the cognate antigen of the OT-I T-cell receptor, chicken ovalbumin$_{257-264}$ peptide presented on murine class I major histocompatibility complex (MHC) (H2Kb-OVA) to trigger cytotoxic response in CTLs. The DAAM-particles elicited antigen-dependent Calcium ($Ca^{2+}$) influx (Supplementary Fig. 12), indicative of TCR-dependent T-cell activation. Since actin reorganization is critical for force generation in the IS[35], CTLs were transduced with LifeAct-eGFP for visualization of actin dynamics.

CTLs induced strong deformation of the DAAM-particles, with a pattern clearly distinct from the deformations observed during phagocytosis (Fig. 6). The circular cell–particle contact area, which was obtained from Lifeact-eGFP imaging was ~8 μm in diameter and deeply indented (~2.5 μm), leading to almost inverted curvature throughout the contact area (Fig. 6a). The contact area contained outward directed shear forces (up to 100 Pa) and was surrounded by a narrow, and slightly protruding rim, which was the result of local pulling (50–100 Pa). Although flattening of T-cell–target synapses has long been observed[36], the surprising crater-like topology we observe may reflect the synapse topology enforced by T cells on soft substrates that do not actively resist T-cell burrowing.

Importantly, these live cell experiments allowed us to track the dynamics of the CTL-target interaction, and over the timescale of minutes we observed the large scale indented area becoming deeper (Fig. 6b), and localized indentations (up to 200 Pa, 0.5 nN total force) forming within the contact area (Fig. 6c). Pulling in the ring was also highly dynamic with discrete spots (up to 100 Pa, 0.2 nN total force) appearing and vanishing within minutes. Finally, correlation of force exertion with LifeAct-eGFP fluorescent signal revealed that these temporal changes in pushing and pulling were accompanied by coincident local increases in actin density.

## Discussion

Here we have introduced batch-produced, deformable, and readily derivatizable microparticles as cellular stress sensors, as well as a numerical method to calculate forces exerted on them by cells with high spatial resolution and sensitivity. Aside from force sensing applications, such particles can serve as ideal model targets for studying cellular processes, such as cell–cell interactions. They provide the unique possibility to mimic the size, rigidity, and relevant chemical characteristics of cells. The tunability of the particles' Young's moduli between 0.3 and 10 kPa covers a significant part of the range exhibited by various cell types[37], and can likely be expanded further. In contrast, polystyrene or silica, which are often used as model targets, have Young's moduli of 2 GPa and 70 GPa, respectively. This is $10^5$–$10^7$ fold larger than cells and most other biological materials. Moreover, for microscopy applications, the small refractive index difference between DAAM-particles and the surrounding medium ensures little optical distortion in imaging of particles themselves, and results in improved visualization of surrounding cellular structures.

Recent reports have described deformable hydrogel particles that can be used as force sensors in multicellular aggregates and tissues[17,18]. Compared to the single microfluidic channel methods chosen previously, our batch particle synthesis method is simpler, and requires little specialized expertise or facilities. In terms of resulting particle properties, our production method leads to good monodispersity (CV ≲ 0.1), approaching that reached with a single microfluidic channel (CV ~ 0.05). Importantly, our observed particle-to-particle variation in optical and mechanical properties is markedly low, and ~10 fold lower than reported previously (Supplementary Discussion). Furthermore, we show uniform protein conjugation to the acrylic acid incorporated in DAAM-particles. Such efficient conjugation is crucial for robust triggering of specific cellular responses and, by allowing homogeneous staining, particle edge localization.

We have illustrated how our approach can be used for recording subcellular mechanical behavior in cell–target interactions in great detail. Critical to achieving the required high spatial resolution and sensitivity is our strategy for measuring microparticle deformation and calculating forces. Unlike classical TFM, which was recently extended to deformable microparticles (~25 μm)[17] and is based on measuring displacement of tracer particles embedded in hydrogels, our approach is based on measurement of the particle boundary with superresolution

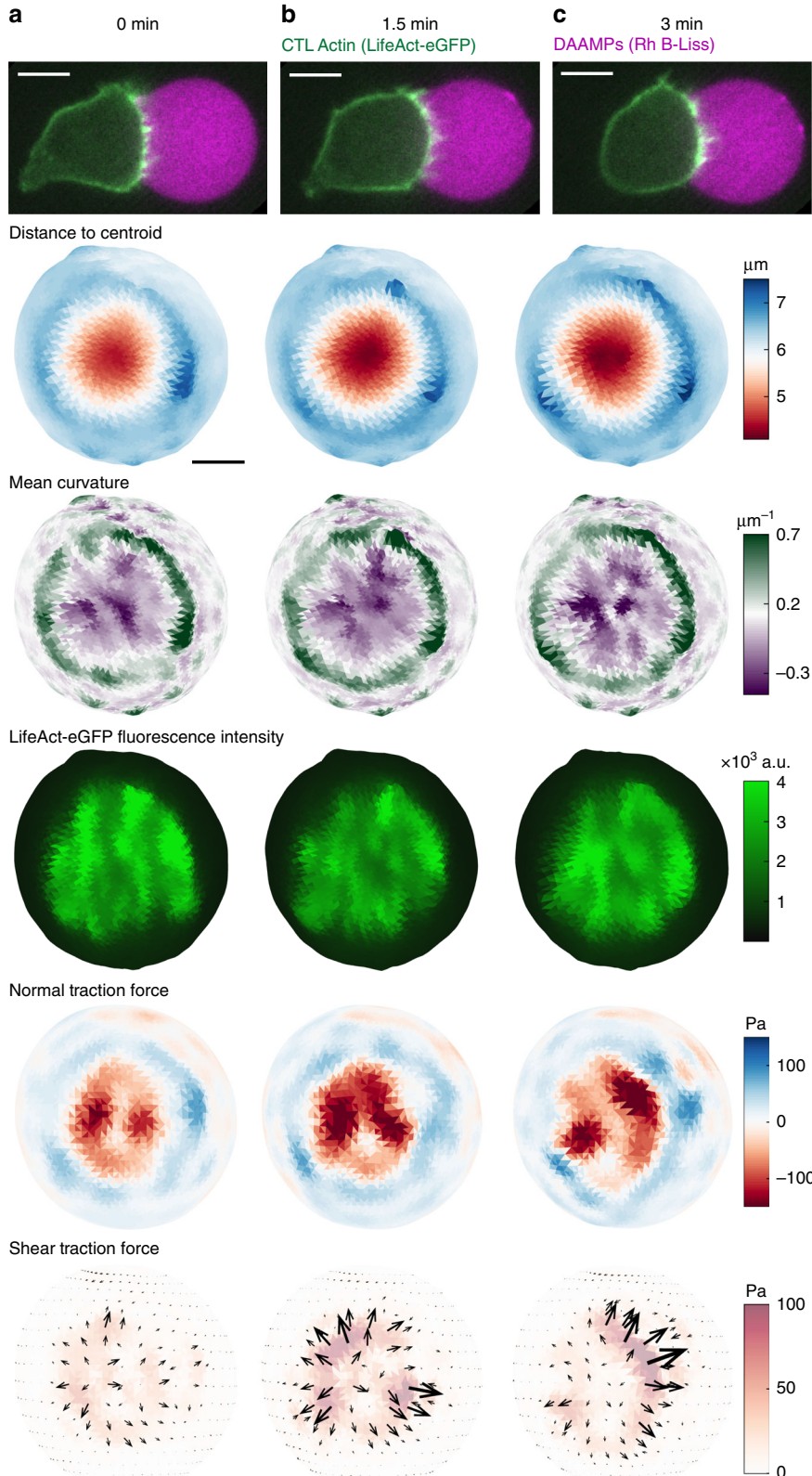

(~50 nm) accuracy (Supplementary Discussion). Although computationally more challenging, the sampling in this strategy is not limited by the tracer-to-tracer spacing (>1 μm) and the approach is sensitive because deformation is measured directly at the point of force application. Like most inverse TFM methods[38], our computational approach is only appropriate for linear elastic materials, and is currently also limited to small strains ($\varepsilon \lesssim 0.1$),

although it could in principle be extended for large strains. Importantly, our method also provides a reference-free estimate of cellular stresses removing the need to separately measure the undeformed particle[39].

We have focused on immune cell–target interactions and demonstrated detailed quantification of patterns of cellular force exertion during these processes. Moreover, we have shown that

**Fig. 6 Live cell force exertion and model-target deformation during the T-cell cytotoxic response. a–c** Frames from a time-lapse of the mechanical activity in the cytotoxic T-cell (CTL) immunoligcal synapse measured with a soft ($C_c$ 0.32%, $E_y$ 0.3 kPa) DAAM-particle functionalized with Rhodamine B-Lissamine (Rh B-Liss), ICAM−1 and OVA$_{257-264}$ presented on MHC1. Top row, iSIM images (z-slices) of a CTL transfected with LifeAct-eGFP (green) triggered by a DAAM-particle (magenta). 2nd row, CTL-induced shape change of the DAAM-particle and, 3rd row, resulting particle mean surface curvature. 4th row, summed radial intensity projection of CTL actin distribution (LifeAct-eGFP signal) within 1 μm of the particle surface. Bottom 2 rows, estimated normal and shear traction forces exerted by the CTL. Visualization of distance to centroid, mean curvature and forces are similar to Figs. 4 and 5. Images are representative examples from multiple ($n > 50$) independent experiments. Color scale legends are presented above each row on the left. White scale bars are 5 μm; black scale bar is 3 μm.

the dynamics of such processes can be captured using live cells and time-lapse microscopy, and can be correlated with cellular protein distributions. As also exemplified by the development of an analogous approach for quantifying forces during development in zebrafish embryos by Träber et al.[40], the method presented here is expected to be broadly applicable in the study of inter- and intracellular forces in vitro and in vivo.

## Methods

**Synthesis of deformable acrylamide acrylic acid particles**. All reagents were reagent grade and purchased from major chemical suppliers unless otherwise specified. All acrylamide mixtures contained 150 mM NaOH, 0.3% (v/v) tetra-methylethylenediamine (TEMED), 150 mM MOPS (prepared from MOPS sodium salt, pH 7.4) in addition to the acrylamide (AAm), acrylic acid (AAc) and cross-linker N,N'-methylenebisacrylamide (BIS) with a final pH of 7.4. The total mass concentration of acrylic components ($C_T = C_{AAm} + C_{AAc} + C_{BIS}$) was 100 mg/mL, and the relative concentration of acrylic acid ($m_{AAc}/(m_{AAm} + m_{AAc} + m_{BIS})$) was 10% for all mixtures. The crosslinker concentration ($C_c = m_{BIS}/(m_{AAm} + m_{AAc} + m_{BIS})$) was varied between 0.32% and 2.3%, as indicated throughout the text and figures. Prior to extrusion, the mixture was degassed for 15 min and kept under nitrogen until the extrusion process was complete. Tubular hydrophobic Shirasu porous glass (SPG) membranes of 20 mm length, 10 mm outer diameter and with pore size (diameter) 0.5–3 μm, (SPG Technology) were sonicated under vacuum in HPLC grade n-heptane to remove gas trapped in the membrane. The membrane was mounted on an internal pressure micro kit extruder (SPG Technology) and immersed into an oil phase (~125 mL) consisting of hexanes (99% ACS reagent, mixed isomers) and 3% (v/v) Span 80 (Fluka, 85548). 10 mL of gel mixture was extruded through SPG membranes under nitrogen pressure of ~7 kPa, 8 kPa, 30 kPa and 100 kPa, for membranes with pore size 3 μm, 1.9 μm, 1.1 μm and 0.5 μm, respectively. The oil phase was continuously stirred at 300 rpm and kept under nitrogen atmosphere in a 3-neck water-jacketed flask (Fig. 1a, custom made by Chemglass Life Sciences), during extrusion (and polymerization). After completion of extrusion, the emulsion temperature was increased to 60 °C. Once the temperature equilibrated, DAAM-particle polymerization was induced by addition of ~225 mg granular 2,2′-Azobisisobutyronitrile (AIBN) (1.5 mg/mL final concentration). The polymerization reaction was continued for 3 h at 60 °C and then at 40 °C overnight. Polymerized particles were subsequently washed (5× in hexanes, 1× in ethanol), dried under nitrogen flow for ~30 min, and resuspended in PBS (137 NaCl, 2.7 mM KCl, 8.0 mM Na$_2$HPO$_4$, 1.47 mM KH$_2$PO$_4$, pH 7.4).

**Microparticle functionalization**. DAAM-particles were diluted to 5% (v/v) concentration and washed twice in activation buffer (100 mM MES, 200 mM NaCl, pH 6.0). They were then incubated for 15 min in activation buffer supplemented with 40 mg/mL 1-ethyl-3-(3-dimethylaminopropyl)carbodiimide, 20 mg/mL N-hydroxysuccinimide (NHS) and 0.1% (v/v) Tween 20. Afterwards they were spun down (2 min, 16,000g) and washed 4× by centrifugation in PBS8 (PBS adjusted to pH 8.0 with NaOH) with 0.1% Tween 20. Immediately after the final wash the particles were resuspended in PBS8 with BSA (Sigma, A3059) (or FITC-BSA for the experiment in Fig. 1e and Supplementary Fig. 2) (for concentrations see Supplementary Table 1). After 1 h the cadaverine-conjugate was added: either Alexa Fluor-488 Cadaverine for the figures illustrating the lensing effect (Fig. 1i, Supplementary Fig. 5) or Tetramethylrhodamine Cadaverine for all other experiments (Supplementary Table 1). After 0.5 h unreacted NHS groups were blocked with a solution of: 100 mM TRIS and 100 mM ethanolamine (pH 9). Finally, DAAM-particles were spun down (30 s, 8000g) and washed 4× in PBS. DAAM-particles were further functionalized with anti-BSA rabbit IgG (MP Biomedicals, 0865111) in PBS (for concentrations see Supplementary Table 1) and spun down (30 s, 8000g) and washed 3× with PBS for phagocytic assays and lensing effect experiments. For the lensing effect experiments, DAAM-particles were resuspended in 4.5 μg/mL Alexa Fluor 546 goat anti-rabbit IgG in PBS, again followed by 3 spins and washes (PBS). 10 μm yellow-green fluorescent carboxylate modified polystyrene beads (Polysciences) were functionalized with BSA, anti-BSA rabbit IgG, and Alexa Fluor 546 goat anti-rabbit IgG similarly (Supplementary Table 1).

For T-cell experiments, DAAM-particles were functionalized similarly to above. Approximately $4.1 \times 10^7$ particles (estimated by haemocytometer) were conjugated

with 9.0 μM streptavidin (Prozyme, SA10). After 1 h of conjugation with protein, 10 μM Lissamine Rhodamine B-ethylenediamine (Thermo Fisher Scientific, L2424) was added. After 0.5 h blocking and washing was performed as above. On the day before imaging, particles were spun down (6000g, 2 min) and washed twice with PBS, and incubated with biotinylated antigens and adhesion molecules at room temperature for 2 h while mixing. The antigen complex used was 0.43 μg/mL purified biotinylated H2-K$^b$-OVA (chicken ovalbumin (OVA) residues 257–264 presented on murine class I major histocompatibility complex (MHC) molecule H2-K$^b$), and the adhesion molecule used was 26.1 μg/mL purified biotinylated intercellular adhesion molecule 1 (ICAM-1). This antigen concentration was chosen based on the saturation of CD69 expression after incubation with functionalized DAAM-particles (Supplementary Fig. 12). After incubation, the particles were spun down and washed thrice with PBS.

**Fluorescent conjugation of BSA**. Fluorescein isothiocyanate conjugated BSA (FITC-BSA) was made according to previously published techniques[41]. Briefly, 800 μl of 5 mg/mL FITC in DMSO was added to 100 mg of BSA in 10 mL of carbonate buffer (pH 9.0). After 0.5 h of stirring at room temperature the reaction was continued at 4 °C overnight. The following day the labeling reaction was quenched with 1 mL of 1 M Tris HCL (pH 8.8) followed by dialysis against PBS for 2 h at 4 °C. The dialysate was concentrated to 1.5 mL by placing the dialysis bag in solid sucrose, and desalted over a G25 column equilibrated with PBS.

**Microparticle size measurements**. For microparticle size measurements, particles were imaged with a 20× NA 0.4 objective using phase-contrast. For the smallest particles, which were extruded through 0.5 μm pore size SPG filters, a 40× NA 1.3 objective was used. Images were analyzed in ImageJ, by smoothing (2×), performing edge detection, binarizing images by thresholding and filling holes. Finally, adjacent particles were separated using the watershed algorithm, and particle radii were estimated from particle areas.

**Microparticle refractive index measurements**. 3D quantitative refractive index (RI) maps were measured on a 3D Cell Explorer (NanoLive) and analyzed using ImageJ. Z-slices of the 3D tomogram through the center of individual DAAM-particles were selected for analysis. Histograms of the RI in a small region of interest around the DAAM-particle revealed a clear bimodal distribution. $RI_{MP} - RI_{medium}$ was then measured as the distance between the two peaks of the RI histogram. $RI_{medium}$ was measured for PBS (1.334), or used as specified by the manufacturer as 1.45 for Vectashield mounting medium (Vectorlabs, H-1000).

**Atomic force microscopy measurements**. Unfunctionalized DAAM-particles and BSA + Cad-DAAM-particles were attached to poly-L-lysine coated and untreated circular 25 mm glass coverslips, respectively. All coverslips were cleaned for 5 min in detergent Micro-90, rinsed with milliQ water and dried at 200 °C. For poly-L-lysine coating, coverslips were then incubated in a 0.1 mg/mL poly-L-lysine (Sigma, P8920) solution for 1 h, rinsed with water and left to dry at room temperature. 50 μl of DAAM-particle suspension (~2% solids (v/v)) in PBS was added to the coverslips and subsequently washed 3 times with PBS. AFM experiments were performed with large radius (~600 nm) pyramidal tips (LRCH-15-750-R, Nanoscience Instruments) on a Bruker Bioscope Resolve setup. The spring constant of individual cantilevers (~0.13 N/m) was calibrated by thermal tuning. Before indentation, particles were imaged in peak force tapping mode (peak force setpoint 0.5–1 nN). Subsequently, 2–5 nanoindentations were performed on the center of particles. AFM data was analyzed using custom Matlab software. DAAM-particle shape was determined as described previously for small vesicles[42]. Briefly, a spherical arc was fit to upper part of the DAAM-particle (exceeding half its height) to obtain the radius of curvature $R_{MP}$, which was corrected for tip convolution by subtracting the tip radius ($R_{tip}$). Assuming that adherent particles form spherical caps, the shape was then approximated from $R_{MP}$ and the particle height (Supplementary Fig. 6). Indentation curves (or force deformation curves, FDCs) were first processed by correction for drift in the baseline and subtraction of a fit to the cantilever response, which was obtained by performing indentations on bare glass surfaces. The resulting DAAM-particle indentation curves were analyzed by fitting a Hertzian model up to 0.5 $R_{tip}$, with the contact point as a fitting parameter. For these fits, the effective radius $R^*$, where ($1/R^* = 1/R_{tip} + 1/R_{MP}$), was used. FDCs for the stiffer DAAM-particles ($C_c$ 0.65%

and $C_c$ 2.3%) were analyzed above 50 pN force. Due to the low rigidity of the softest DAAM-particles ($C_c$ 0.32%) a force threshold of 10 pN was used. The Young's moduli ($E_y$) obtained from each of the 2–5 FDCs for individual particle were averaged to reduce the measurement error per DAAM-particle. For the Young's moduli calculations, DAAM-particles were assumed to be incompressible (Poisson's ratio = 0.5), which was previously shown to be a good approximation for bulk[43] and microparticle[18] acrylamide gels. Furthermore, the indentation-induced deformation was assumed to only deform the upper part of the DAAM-particles, which is in contact with the AFM tip. The significant deformation of the particles due to surface adhesion and the resulting large contact radius (>4 μm) with the surface (Supplementary Fig. 6), renders the indentation-induced deformation of the bottom of the DAAM-particles, which is in contact with the glass surface, negligible compared to the deformation of the upper part. For bulk gel measurements, hydrogels, covalently bound to glass substrates, were prepared as described under the section PSF measurements, but with compositions, and buffer conditions, identical to the DAAM-particles (for all three crosslinker concentrations). Measurements were performed with spherical tips ($SiO_2$) with ~300 nm radius (NovaScan Technologies). Indentation curves were analyzed as described for the DAAM-particles.

**Bulk modulus measurements**. Bulk modulus measurements were performed as described previously[40,44]. Briefly, dextran solutions with known osmotic pressure[45] were made from FITC-conjugated dextran (molecular weight: $2 \times 10^6$ g/mol, MilliporeSigma 52471). Particles were imaged using confocal microscopy and their volume was estimated from the maximum surface area in the confocal image stack.

**Cell culture**. Murine macrophage-like cells J774A.1 (ATCC, TIB-67) were maintained and subcultured according to the methods recommended by ATCC. Briefly, cells were maintained in DMEM medium (Gibco, 11965–092) supplemented with 10% FBS (GemBio, 900–108), and antibiotics and antimycotic (Gibco, 15240–062). Subcultures were prepared by scraping.

Cultured cytotoxic T lymphocytes (CTLs) were obtained from Rag2/OT-I transgenic mice (Taconic). The animal protocols used for this study were approved by the Institutional Animal Care and Use Committee of Memorial Sloan-Kettering Cancer Center. Briefly, splenocytes from C57BL/6J wild-type mice were pulsed for 2 h with 100 nM OVA 257–264 at 37 °C, washed to remove excess peptide, and then mixed with Rag2/OT-I splenocytes for 2 days. On the second day, activated lymphoblasts were purified by density gradient centrifugation and retrovirally transduced. Cells containing the transgene (Lifeact-eGFP) were then selected for 1 day and maintained until imaging on day 7. All cultures were maintained in RPMI medium containing 10% FBS (VWR, 97068–085) and 30 IU/mL human interleukin-2 (hIL-2) (Roche).

**Retroviral transfection**. Phoenix E cells were first transfected with expression vectors for Lifeact-eGFP together with packaging plasmids via the calcium phosphate method on the day of spleen harvesting. Ecotropic viral supernatants were then collected after 48 h at 37 °C, filtered through a 0.20 μm sieve, and added to $1.5 \times 10^6$ lymphoblasts on day 2. These mixtures were centrifuged at 1400$g$ in the presence of polybrene (4 μg/mL) at 35 °C for 2 h. Cells were then split 1:3 in complete RPMI, and then grown, selected, and maintained as stated above until day 7.

**Phagocytic assays**. For phagocytic assays, cells were transferred to 12-well glass bottom plates (Cellvis, P12–1.5H-N) ($1.5 \times 10^5$ cells/well). 1 h before addition of DAAM-particles, the medium was replaced by L-15 medium without serum. 15 min before addition of the DAAM-particles Hoechst 33258 (Thermo Fisher Scientific, H3569) was added for a final concentration of 1 μg/mL in each well. The medium was then replaced with 200 μL L-15 per well including the DAAM-particles ($7.5 \times 10^5$ DAAM-particles/well), the plate was spun for 1 min at 300$g$ and incubated at 37 °C. Cells where fixed by direct addition of 1 mL of 4% formaldehyde (J.T. Baker, 2106–01) in PBS after 5 min for the phagocytic deformation assay (Figs. 3–5) and after 30 min for the uptake efficiency assay (Supplementary Fig. 8). Cells were rinsed vigorously by pipetting up and down with PBS to remove particles not in contact with cells. Exposed surface of the particles was immunostained with 4 μg/mL Alexa Fluor-647 donkey anti-rabbit IgG (Thermo Fisher Scientific, A31573) in PBS. After 30 min the wells were rinsed with PBS (3×) and imaged on a confocal microscope.

**Cytotoxic T lymphocyte assays**. A #1.0 8-well coverglass chamber (Thermo Fisher Scientific, 155411) was coated with biotinylated poly-L-lysine in water for 0.5 h, blocked for 1 h with 20 mg/mL BSA in HBS, and then washed extensively with PBS. To each well, $2 \times 10^5$ DAAM-particles in PBS were added and left to settle overnight at 4 °C. On the day of imaging, each well was washed gently but extensively with PBS to remove non-adherent particles, and then replaced with imaging medium (colorless RPMI with 5% FBS and 30 IU/mL hIL-2). Cells were washed twice with imaging medium to remove traces of phenol red from the original culture medium, and then incubated at 37 °C. Glass chambers containing DAAM-particles were first allowed to thermally equilibrate on the microscope stage at 37 °C. Then, planes/regions of interest containing evenly spaced DAAM-

particles were selected by scanning across a given well at low laser power. Up to $2 \times 10^5$ cells were pipetted into the well and allowed to settle for ~2 min Once a sufficiently bright T-cell–DAAM-particle conjugate of interest was identified, time series image stacks were acquired on an instantaneous structured illumination microscopy (iSIM) system.

For measurement of $Ca^{2+}$ influx in T cells, chamber slides with peptide-coated DAAM-particles (8.6 μm, 4.0 kPa) were prepared in a similar manner to the above. CTLs were loaded with 5 μg/mL Fura-2AM, washed, and then added to the wells. Brightfield, RFP, 340 nm, and 380 nm images were acquired every 10 s for ~35 min Population $Ca^{2+}$ responses were analyzed using Slidebook software (3i) as follows: after image background correction, masks were generated using Otsu thresholding and size exclusion (8-μm minimum). Then, T-cell events that overlapped with DAAM-particle events for more than 2 frames (20 s) were selected and the 340/380 intensity ratios were normalized using the baseline value before the initial rise.

**Microscopy**. 3D confocal imaging was performed using a spinning disk CSU-W1 scanner unit (Yokogawa) and a DMi6 inverted microscope, equipped with a 100× NA 1.40 objective (Leica Microsystems). Excitation was done with solid state lasers (100 mW 488 nm, 50 mW 560 nm, 100 mW 642 nm) and images were captured on an Ixon Ultra EMCCD camera (Andor). Z-stacks with 100 nm step size were recorded using a piezo stage insert (LUDL Electronic Products, 96A900). The setup was operated using MicroManager[46]. Edge localization precision and reference shape measurements were performed with the 50 mW 560 nm laser, where laser power output was varied between 1–50% and image stacks were acquired with 50 ms exposure per slice (144 μm × 144 μm each).

Instantaneous structured illumination microscopy (iSIM) was performed using a VisiTech iSIM scan head (VisiTech International), an Optospin fast filter wheel (Cairn), and an IX73 microscope, equipped with a 60 × NA 1.3 silicone oil objective (Olympus) and a Hamamatsu Flash 4.0 v2 camera. For live cell imaging a WSKM-F1 stagetop incubator (Tokai Hit) was used. Images were acquired using MetaMorph software (Molecular Devices).

**Microparticle 3D shape reconstruction**. Image analysis was performed with custom software in Matlab. Individual microparticles were segmented based on intensity thresholding confocal z-stacks, using the dip in the distribution of the logarithm of voxel intensities as the threshold. The centroid of DAAM-particles was estimated, and they were isolated in ~20 × 20 × 30 μm regions of interest (ROIs) that were used for further processing. First, images were deconvolved with a point spread function (PSF), measured ~8 μm from the sample surface (Supplementary Fig. 4). Deconvolution was performed with 25 iteration steps using a Landweber algorithm, and was implemented using DecovolutionLab 2[47]. Background subtraction was performed by averaging the Z-slices outside the ROI along the axial direction and subtraction of the resulting 2D image from each slice of the ROI. At this point, cubic interpolation along the z-axis was used to equalize the voxel dimensions. Edge detection was performed using the 3D Sobel operator[48] (Supplementary Fig. 7), after which the total edge magnitude was calculated as the root-mean squared of the three resulting images. For superlocalization of edge coordinates, first a volume estimate of each DAAM-particle was made, based on the number of thresholded voxels, which was converted into a particle surface area estimate assuming a perfect sphere. The number of edge coordinates $N$ was determined as the particle surface area divided by the area of a circle with radius 250 nm (for the DAAM-particles subjected to phagocytosis) or 500 nm (for undeformed spheres, in which case we wanted to prevent correlation between neighboring points). Longitude $\varphi$ and co-latitude $\theta$ for approximately equally spaced points were found as: $\varphi = n\pi(3 - \sqrt{5}) \bmod 2\pi$, $\theta = \arccos(1 - 2n/(N - 1))$, where $n = 0 \dots N - 1$. Modulus $2\pi$ is taken, such that $\varphi \in [0, 2\pi)$. Cubic interpolation was used to estimate the intensity values along lines originating from the particle centroid in the directions of the angles determined above. Then a Gaussian function was fit to each line profile to determine the location of the edge with subpixel precision.

**Smoothing of edge coordinates**. Edge coordinates were smoothed (for sphericity, curvature and force calculations) using an equivalent of a 2D moving average, operating on the radial component of edge coordinates. Great circle distances ($d$) between edge coordinates with indices $i$ and $j$ were calculated along a perfect sphere: $d = \arccos\left(\sin\theta_i \sin\theta_j + \cos\theta_i \cos\theta_j \cos\left(\varphi_i - \varphi_j\right)\right)R$, where $R$ is the equivalent diameter of a sphere to the particle. The radial component of the edge coordinates was then averaged within the given window size (1 μm²).

**Calculation of particle properties**. Calculation of particle properties was performed using custom Matlab software. A triangulation was generated between points with the same angular coordinates (longitude and latitude) as the particle edge coordinates, but on a unit sphere. Then a convex hull approach was used to connect all data points and not leave any gaps. The particle surface area ($S$) was then calculated as the sum of the surface areas of all triangles. The particle volume ($V$) was calculated as the sum of the (signed) volumes of tetrahedra, which were formed by connecting all triangles with an arbitrary point. The particle centroid was determined as the volume-weighted sum of the tetrahedra centroids. Before calculation of sphericity and surface curvature, the radial component of the edge

coordinates was first smoothed within a 1 μm$^2$ window (see above). This was done because of the high sensitivity of these measures to high-frequency noise. Sphericity was calculated as $\Psi = (6\pi^{1/3}V^{2/3}S^{-1})/2^{1/3}$, where the division by $2^{1/3}$ was used when only the upper hemisphere was analyzed, as it is a correction that guarantees that $\Psi$ cannot exceed 1 for non-closed shapes[49]. For surface curvature calculations, first principal curvatures ($k_1$ and $k_2$) of the triangulated mesh were determined as described previously[50,51]. Finally, mean curvature $H = (k_1 + k_2)/2$ and Gaussian curvature $K = k_1 k_2$ were calculated. Local root-mean-squared surface roughness ($R_{rms}$) measurements were performed by using great circle distances to find points within a 1 μm$^2$ area (as above, typically 6–8 points), and then taking the standard deviation of the radial component of these points. Global $R_{rms}$ was calculated similarly, but over the entire upper hemisphere.

**Stress-free boundary analysis.** Analysis of the immunostaining of the free particle surface was performed in custom Matlab software (Supplementary Fig. 7). First, images were deconvolved as described above. Then, cubic interpolation was used to estimate the intensity values along lines originating from the particle centroid (as determined during particle shape analysis). A regular grid in spherical coordinates (as opposed to an approximately equidistant grid in particle shape analysis) was used, to simplify later image processing steps (in particular the active contour algorithm described below). The number of points in the regular grid was approximately equal to the number of points used for shape analysis. Generated line profiles revealed a Gaussian intensity function, and the integrated intensity $I_{tot}$ under the Gaussian was determined. $I_{tot}$ was approximated as $1.065 \times I_{max} \times$ FWHM, where $I_{max}$ is the maximum fluorescent intensity, FWHM is the full width at half maximum of the Gaussian peak in the intensity profile and 1.065 is the prefactor that appears during integration. $I_{tot}$ appeared more uniform over the DAAM-particle surface than the $I_{max}$, because of the non-isotropic resolution in confocal microscopy. After generating a regular grid with total intensity values, this signal was binarized, initially by using a dip in the distribution of the logarithm of pixel intensities as the threshold. Then, a region-based active contour (or snake) algorithm[52] was used to optimize the mask. Finally, interpolation of the mask was used to determine for each particle edge coordinate if it was part of the freely exposed surface.

For live cytotoxic T-cell assays, the LifeAct-eGFP fluorescence signal was used for determination of the cell-contact area. However, radial line profiles of the fluorescent signal was not approximated by a Gaussian function, and instead the pixel intensities up to 1 μm from the particle edge were summed.

**Reference-free estimation of stresses from particle shape.** Force calculations were performed with custom Python package ShElastic[19]. The particle shapes were smoothed within a 1 μm$^2$ window, and the edge of the traction-free boundary was dilated (Supplementary Note 4). Resolving traction forces from the particle shape can be formulated as a linear elasticity problem on the sphere $\Omega$ subjected to the constraints that the deformed shape matches the measured shape and that the traction force is zero on the traction-free region. For small-strain linear elasticity problems, the equilibrium condition (based on generalized Hooke's law) of an isotropic elastic continuum $\Omega$ can be expressed in terms of the displacement field $\mathbf{u}$ as:

$$\mu\nabla^2\mathbf{u} + \left(\frac{\mu}{1-2\nu}\right)\nabla(\nabla \cdot \mathbf{u}) = 0, \tag{2}$$

where $\mu$ is the shear modulus and $\nu$ the Poisson's ratio. This equation corresponds to the condition of static equilibrium in the absence of body forces, which are likely negligible compared to total surface traction forces. The deformed shape of the sphere corresponding to any displacement field $\mathbf{u}$ can be represented as a radial function $r(\theta, \varphi)$, which is the distance between the origin and the deformed surface. Let $r^s(\theta, \varphi)$ be the radial function of the measured 3D shape. Hence the constraints on the elasticity problem can be written as $r(\theta, \varphi) = r^s(\theta, \varphi)$, $\mathbf{T}|_{\partial\Omega_t} = 0$.

We approach this problem by solving a series of elasticity problems on the sphere, each subjected to a different surface displacement field that satisfies the measured shape exactly. The surface displacement field is adjusted iteratively to reduce the traction on the traction-free region. The surface displacement field is sampled on a Gauss-Legendre quadratic (GLQ) mesh $(\theta_0, \varphi_0)_i$, $i = 1 \dots N$. The surface displacement field $\mathbf{u}(\theta,\varphi)$ can be represented on every mesh node in spherical coordinates as $(\Delta r, \Delta\theta, \Delta\varphi)_i$, where $(\Delta\theta, \Delta\varphi)_i$ are our fundamental degrees of freedom, and $\Delta r_i$ are obtained by projecting the mesh nodes onto the measured shape in the deformed state.

Given the trial surface displacement field $\mathbf{u}(\theta, \varphi)$, we first transform it to Cartesian coordinates, $u_j(\theta, \varphi), j = x, y, z$, and then expand it into spherical harmonic space up to degree $l \le l_{max}$:

$$u_j(\theta, \varphi) = \sum_{l=0}^{l_{max}} \sum_{m=-l}^{l} \hat{u}_{jlm} Y_l^m(\theta, \varphi), \tag{3}$$

where $\hat{u}_{jlm}$ are coefficients, and $Y_l^m$ is the spherical harmonic function defined in Supplementary Note 5. We then use the spherical harmonics method[19] to obtain the corresponding surface traction field $\mathbf{T}(\theta, \varphi)$ that satisfies the equilibrium condition (2). In this work $l_{max} = 20$, which was chosen as a good balance between

lateral spatial resolution (~750 nm) and computational time (~0.007 s per traction evaluation). The surface traction field is also expressed in the spherical harmonic space:

$$T_j(\theta, \varphi) = \sum_{l=0}^{l_{max}} \sum_{m=-l}^{l} \hat{T}_{jlm} Y_l^m(\theta, \varphi), \tag{4}$$

where $\hat{T}_{jlm}$ are coefficients that are completely determined by $\hat{u}_{jlm}$.

We define a measure of residual traction magnitude on the traction-free region as $R(\mathbf{T}; \partial\Omega_t)$. Therefore, our goal is to determine $(\Delta\theta, \Delta\varphi)_i$ that minimizes $R(\mathbf{T}; \partial\Omega_t)$. However, the solution of such a problem is not necessarily unique. Therefore, we solve an alternative optimization problem:

$$\min_{\{\mathbf{u}_s\}} f(\mathbf{u}_s), \tag{5}$$

$$f(\mathbf{u}_s) = E_{el} + \alpha R^2(\mathbf{T}; \partial\Omega_t) + \beta E_{pen}(\mathbf{T}), \tag{6}$$

where the cost function $f$ has contributions from: (1) the elastic energy $E_{el}$; (2) the traction residual $R(\mathbf{T}; \partial\Omega_t)$; (3) an anti-aliasing filter (Supplementary Note 3). The elastic energy is included to favor lower energy solutions, which are likely to be more realistic. Since $R$, $E_{el}$, $E_{pen}$ all have different units, the parameters $\alpha$ and $\beta$ have the appropriate units to make $f$ have the units of pJ. The selection of appropriate $\alpha$ and $\beta$ values is illustrated in Supplementary Fig. 10. For minimization of $f(\mathbf{u}_s)$, we used the conjugate-gradient algorithm implemented in SciPy. Convergence was reached after ~5000 iterations, while during each iteration the cost function and its gradient were evaluated ~20 times.

The residual traction magnitude on the traction-free region is calculated as:

$$R(\mathbf{T}; \partial\Omega_t) = \sqrt{\frac{1}{N} \sum_{(\theta_0+\Delta\theta, \varphi_0+\Delta\varphi) \in \partial\Omega_t} \left\| \mathbf{T}_i(\theta_0, \varphi_0) w(\theta_0) \right\|^2}, i = 1 \dots N, \tag{7}$$

where $N = (l_{max} + 1)(2l_{max} + 1)$ is the total number of GLQ mesh nodes and $w(\theta) = \sin\theta$ is the weight function to prevent over-sampling of data points at the poles compared to the equator (Supplementary Note 2). Due to the orthogonality of spherical harmonic functions, the elastic energy can be calculated as a sum over all spherical harmonic modes:

$$E_{el} = 2\pi r_0^2 \sum_{j=0}^{3} \sum_{l=0}^{l_{max}} \sum_{m=-l}^{l} \left[\hat{u}_{jlm}\right]^* \hat{T}_{jlm}, \tag{8}$$

where $[u]^*$ represents the complex conjugate of $u$; See Supplementary Note 5 for derivation.

**Displacement solution based on spherical harmonics.** As we are using an iterative method for solving the elasticity problem, it is essential to have fast computation of the shape $\mathbf{T}(\theta, \varphi)$ from the surface displacement field $\mathbf{u}(\theta, \varphi)$. The spherical harmonics method was previously developed as a fast solver for linear elasticity problems with spherical interfaces[19]. Briefly, the method can solve the stress field $\sigma(r, \theta, \varphi)$ and the displacement field $\mathbf{u}(r, \theta, \varphi)$ everywhere in the elastic medium $\Omega$, given the traction boundary condition $\mathbf{T}_b(\theta, \varphi)$ or the displacement boundary condition $\mathbf{u}_b(\theta, \varphi)$ on the spherical interface $\partial\Omega$. Using fast spherical harmonic transformation as provided by SHTools[53], the spherical harmonics method is more efficient and accurate than other general elasticity problem solvers such as finite element methods (FEM)[19]. The implementation of the method is available online as ShElastic toolbox[19]. In particular, ShElastic toolbox pre-calculates a complete set of fundamental displacement solutions with $\mathbf{u}^{(K)}$ that satisfy the equilibrium Eq. (1):

$$\mathbf{u}^{(K)} = \frac{1}{2\mu}\left[ -4(1-\nu)\mathbf{\psi}^{(K)} + \nabla\left(\mathbf{r} \cdot \mathbf{\psi}^{(K)}\right) \right], \tag{9}$$

where the displacement potential vector $\mathbf{\psi}^{(K)}$ is defined in spherical coordinates as:

$$\mathbf{\psi}^{(K)} = \mathbf{\psi}^{(k,l,m)}(r, \theta, \varphi) = \hat{\mathbf{e}}_k \, r^l \, Y_l^m(\theta, \varphi), \tag{10}$$

where $(K) = (k, l, m)$; $(k = x, y, z; l = 0, 1, 2, \dots; m = 0, 1, \dots, l)$ indicates the 3-value index of the solution. The corresponding stress field $\sigma^{(K)}(r, \theta, \varphi)$ in $\Omega$, and the traction field $\mathbf{T}^{(K)}(\theta, \varphi)$ on $\partial\Omega$ are also pre-calculated. Then, the physical properties can be written as a linear combination of the fundamental solutions:

$$\sigma = \sum_K a_K \sigma^{(K)}, \mathbf{u} = \sum_K a_K \mathbf{u}^{(K)}, \mathbf{T} = \sum_K a_K \mathbf{T}^{(K)}, \tag{11}$$

which are infinite series. In practice, we select a cut-off of $0 \le l \le l_{max}$, and select all the $m$ and $k$ modes within the selected $l$ modes to guarantee the symmetry of the deformation. The solution converges with exponentially decreasing error as $l_{max}$ increases. The solution is searched on the solution space $a_K$ as a vector.

**Confocal voxel size calibration.** Differences in axial (z) versus lateral (x,y) length measurements can arise from imperfect calibration of the microscope stage, and from refractive index mismatch between the sample and the immersion oil/coverglass. Hence, accurate calibration is necessary for proper reconstruction of the

particle shape. First, confocal image stacks of 10 µm yellow-green fluorescent polystyrene beads (Polysciences, 18142) were recorded. Then, maximum intensity axial slices through individual particles were selected, and smoothed with a Gaussian filter (with a standard deviation of 3 pixels). Subsequently, edges of the homogeneously stained microspheres were detected using the 2D Sobel operator. The z-location of the equator, and the diameter of the particle, was determined by finding the horizontal line were the distance between the edges (at opposite sides of the particle) was maximal. Gaussian fits to the edge profiles were made to accurately determine the particle diameter. Next, the edge at the apex of the lower hemisphere of the particle (the south pole) was localized using a similar approach. This difference in z-position of the equatorial plane with the south pole was used to estimate the apparent particle radius in z-direction. This approach, were only the lower hemisphere is used, prevents these measurements to be affected by the lensing effect caused by the particle (Fig. 1i, Supplementary Fig. 5). Finally, the ratio of the particle radius measured in xy and the apparent particle radius in z was determined for multiple particles and averaged. In PBS, we found an axial elongation of $1.100 \pm 0.003$ (standard error of the mean (s.e.m.), $n = 34$). In Vectashield (VS) we did not perform such a correction, since particles did not appear elongated in the axial direction (Fig. 2).

**Point spread function measurements**. For measurements of point spread functions (PSFs), red or yellow-green 100 nm fluorescent beads (Thermo Fisher Scientific, F8801/F8803) were embedded in flat adherent pAAm gels. Such hydrogels were prepared similarly to described previously:[54,55] clean 25 mm glass coverslips were incubated for 0.5 h in 2 M sodium hydroxide. Coverslips were rinsed with distilled water and then incubated in 0.5% 3-Aminopropyltriethoxysilane (APTES) (Sigma, 919-30-2) in 95% ethanol. Coverslips were again rinsed with water and incubated in 0.05% glutaraldehyde for 1h, then rinsed with water and dried at 200 ºC. pAAm hydrogels consisting of acrylamide, and crosslinker N,N'-methylenebisacrylamide were made with $C_T$ ($C_{AAm} + C_{BIS}$) 55 mg/mL, $C_C$ ($C_{BIS}/(C_{AAm} + C_{BIS})$) 1.8% (w/w) in distilled water. $2 \times 10^{-5}$% (v/v) 100 nm fluorescent beads and 0.8% (v/v) TEMED were added to the mixture. 0.6 mg/mL ammonium persulfate (APS) (Fisher Scientific, BP179) was added to initiate gel polymerization. A 3.6 µL drop of the mixture was added to a coverslip and was covered with a 12 mm untreated circular glass coverslip, which was gently pushed down. Polymerization was continued for 20 min at room temperature after which the small coverslip was removed. Gels were kept and imaged in PBS (pH 7.4) or in the mounting medium Vectashield (Vector Laboratories, H-1000). Data was analyzed using custom Matlab software. Only beads that had no neighbors in a region of interest of $5 \times 5 \times 10$ µm ($x \times y \times z$) around them were used for the analysis. After background subtraction, the total fluorescent signal of individual particles was normalized to 1. Particles were centered with subpixel resolution based on their weighted centroid (linear interpolation was used to approximate new pixel values). The position of the coverslip ($z = 0$) was determined based on background fluorescent signal, allowing us the measure the height of each particle with respect to the coverslip. Finally, the fluorescent signal of multiple particles ($n > 50$), with similar distances from the coverslip, was averaged.

**Reporting summary**. Further information on research design is available in the Nature Research Reporting Summary linked to this article.

## Data availability

A reporting summary for this article is available as a Supplementary Information file. The data supporting the findings of this study are available from the corresponding authors on request. The source data underlying Figs. 1b–d, f–h, k, 2b–c, and Supplementary Figs. 2c, 3, 6a–b, and 12a–b are provided as a Source Data file.

## Code availability

The Matlab code for analyzing confocal images and deriving particle shape is publicly available on https://gitlab.com/dvorselen/DAAMparticle_Shape_Analysis. The Python code used for analyzing tractions is publicly available on https://gitlab.com/micronano_public/ShElastic.

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

## Acknowledgements

We thank P. Odermatt for help with the refractive index measurements and R.L.D. Labitigan for insightful discussions. This work was supported by the Howard Hughes Medical Institute (D.V., M.F. and J.T.). D.V. further acknowledges the Cancer Research Institute for support through a CRI Irvington fellowship. This work is partly supported by the US Department of Energy, Office of Basic Energy Science, under project DE-SC0010412 (Y.F. and W.C.). We acknowledge additional support by the U.S. National Institute of Health (R01-AI087644) (M.M.d.J. and M.H.). AFM experiments were performed at the Stanford Cell Sciences Imaging Facility and were supported by the NCRR (S10OD021514-01).

## Author contributions

D.V. and J.A.T. conceived of and designed the study. D.V. performed all particle characterization and phagocytosis experiments, wrote the image analysis software and analyzed the data. J.A.T. supervised the project. M.M.d.J. and P.K.S. performed the T-cell experiments under supervision of M.H. M.J.F. aided with the particle synthesis and conjugation. Y.W. and W.C. developed the elasticity theory and performed the traction force calculations. D.V., Y.W., M.M.d.J., M.J.F., M.H., W.C. and J.A.T. wrote the paper.

## Competing interests

The authors declare no competing interests.
