## [Peer Review File · Nature Communications]

Reviewers' comments:

Reviewer #1 (Remarks to the Author):

In this manuscript, Vorselen et al. describe the fabrication and implementation of a new microparticle-based traction force microscopy approach to measure forces during phagocytic engulfment and immunological synapse formation. The manuscript is well-written, and the data are interesting and high quality. In general, I believe the manuscript to be well suited for publication in Nature Communications; however, I have several concerns regarding the force reconstruction methods that would need to be addressed before I could recommend publication. These, as well as minor comments, are outlined below:

Major Comments:

1) The authors present a new force reconstruction approach to obtain both normal and shear forces from measurements of the deformed particle shape. As the authors note, because they only measure deformations at the surface and in the radial direction the problem of relating these deformations to surface tractions is underdetermined. To overcome this, they present an iterative optimization scheme that weights the residual radial displacement, the integrated traction on the region of the sphere thought to be traction-free, and the total elastic energy. The thought being, that the most correct solution will be lowest energy solution that matches the surface geometry yet doesn't predict substantial tractions on the traction-free region. I have several comments on this approach:

a. As this approach is substantially different from prior methods (which only report normal forces as in Campas et al. 2014 or measure the full displacement field as in Mohagheghian et al. 2018), the authors need to more fully demonstrate that their approach is capable of accurately reconstructing both shear and normal surface tractions and define where the limits of resolution lie. This could be done by simulating the forward problem under a variety of loads (both shear and normal) at various characteristic lengths, convolving the ideal surface with the point spread function of their instrument and overlaying Poisson noise and pixelation to simulate the imaging process, and then reconstructing using their standard protocol. This general approach has been outlined in prior studies (e.g. Sabass et al. Biophys J 2008, Legant et al PNAS 2012, Mohagheghian et al. Nature Communications 2018, Colin-York et al. Nanoletters 2016).

b. Does the author's approach require the presence of a traction-free interface? E.G. could it still compute tractions for a fully engulfed microparticle?

c. Is the current approach limited to small-strains or can it be adapted to large strain/non-linearities? Some discussion on the limitations and applicable conditions would be helpful.

d. The 10Pa sensitivity claim in the abstract and discussion seems misleading to me. If I understand correctly, the 10 Pa value is derived from the residual tractions predicted on "traction-free" portion of the microparticle. However, as the authors note in their methods, this surface has been pre-smoothed by a user defined value to "prevent overestimation of high-frequency contributions" and further suppressed by the choice of alpha in the regularization scheme. It seems that this value could be made arbitrarily large or small by the choice of smoothing and regularization parameters. Moreover, the traction sensitivity in loaded regions will depend on the characteristic length of the loading and not simply on the traction amplitude (e.g. two opposing forces in close proximity would produce minimal surface displacement when averaged over a given resolution element and thus may not be detected regardless of their amplitude). In short, force sensitivity and traction length scale are inherently linked in TFM.

e. Several other metrics are commonly used to assess solution accuracy including deviation from static (and rotational) equilibrium, but these aren't discussed here. Are these conditions enforced a-priori by the solution process or will the solutions deviate from these conditions (if so, how

much)? In particular, the shear force calculations (e.g. Fig 5 c, second row, Fig 6) seem to show vortices of shear force that, if accurate, would exert a net rotational moment and cause the microparticle to rapidly spin. Intuitively, this does not seem correct. I also have trouble imagining how such patterns could be generated by the cell.

Minor Comments:

- 1) The super resolution claim in the title is a bit of a stretch. The authors use a similar Gaussian fitting procedure used in Campas et al. (compare Sup Fig 6 from this manuscript to Sup Fig 2 from Campas) to fit the microparticle surface with a precision below the conventional diffraction limit. More generally, the majority of TFM methods fit bead images by fitting a 2D or 3D Gaussian function to determine the bead centroid to well below the diffraction limit and thus "super-resolve" the displacement field. In this sense, nearly all TFM papers could be called "super-resolved" TFM.
- 2) The introduction states "deformable microparticles have so far only been used as passive force reporters without biologically specific ligands". Both Campas et al and Mohagheghian et al used microparticles functionalized with RGD. Campas also functionalized particles with cadherins. I would consider these to be biologically specific ligands.
- 3) To assess particle homogeneity and mechanical uniformity, the authors imaged the radial distributions of conjugated fluorophores. A more rigorous control for isotropic/homogeneous mechanical properties would be to osmotically swell/shrink the particles and analyze the deviation from the starting geometry.
- 4) It would be amazing to see the traction dynamics during a live-cell movie either of phagocytosis or synapse formation. Fig 6 shows only three frames, all of which are already after the synapse is formed. Is there some technical limitation preventing this? Perhaps due to the computational resources required for traction calculation at many different time points? If so, this should be addressed in the discussion.

Reviewer #2 (Remarks to the Author):

The authors report on a new method to measure cell forces through the deformation of soft hydrogel particles. The wider context of this study is traction force microscopy, which traditionally is performed on planar soft elastic substrates. Using beads rather than thick substrates is challenging but recently has been tackled by several groups. Motivated by their earlier Listeria assay for soft particles, the Otger Campas group has injected oil droplets into tissue and monitored its deformations (Refs. 13 and 14). Very recently, the Ning Wang group has used alginate hydrogel beads made in a microfluidic device to measure forces in tumor colonies (Ref. 15). A similar approach is described in a BioRxiv-paper by the Jochen Guck group (Ref. 36), which injected large (17 μm diameter polyacrylamide beads) into zebrafish.

The present work is similar its aims, but has several new elements that make it very appealing to the general audience interested in cell forces. (1) The authors do not use microfluidics (like in Refs. 15 and 36) to generate the particles, but extrusion through porous glass. This leads to relatively monodisperse particles in a size range from 4 and 15 μm . For many groups this approach might be easier to use than microfluidics. The rigidity range is similar as in the other studies, around kPa. (2) Great care is taken to determine particle shape very accurately, with sub-pixel ("super") resolution. This also includes careful measurement of the refractive index. (3) A new computational method has been developed to infer traction stresses. In contrast to Refs. 15 and 36, the authors do not reconstruct the displacement field (either by bead tracking like in Ref. 15 or by a simple geometrical construction like in Ref. 36), but work only with the shape reconstructed in (2). Because shape is not a unique predictor for the tractions, they define a cost function that is minimized in an iterative procedure. Here they use spherical harmonics as published in Ref. 17 by a subset of the present authors. This procedure seems to be impressive and rather rigorous, and sets this work clearly apart from the preprint Ref. 36. (4) Finally the method is applied to two situations of large biological importance, phagocytosis and the

immunological synapse. In both cases, deformation and traction maps are presented and described.

Overall, this study is very solid and impressive, and a clear advance in pushing traction force microscopy into new territories. Its main strength is on the methods side, but there are also very interesting biological results contained in the two examples. In my view, this work is of large interest to the interdisciplinary and general audience of Nat Comms. I do have a few comments on how to further improve the quality and impact of this work.

The Young's modulus is measured by AFM-experiments evaluated with the Hertz-model. I have two comments. First an independent measurement would be good. In Ref. 36, the authors compressed the beads by soluble dextran pressure and showed that the resulting bulk modulus together with the value of the Poisson ratio corresponds to the Young's modulus. I suggest to repeat this procedure here. Second I wonder if there is a possible role of surface tension in this system. Obviously there is a surface tension in the oil phase which gives the spherical shapes, but I am more concerned with the increase in modulus after BSA-functionalization. Does the BSA enter the beads? If it stays at the interface, why should it change the modulus? Should one not extend the Hertzian analysis by a surface tension term?

The biological results are very interesting, but somehow phenomenological. Why do the authors not present the results of actin and/or myosin inhibitions to prove which machinery is behind these forces (like Latrunculin A in Ref. 15)? I also wonder why actin imaging has only been used for the immunological synapse and not for the phagocytosis? Finally I think that the shear force patterns presented in Figs. 5 and 6 should be interpreted. Why do they both show vertex patterns? It is clear from the hairy ball theorem that some singularities have to exist, but why should there be chirality in the problem? Is this something enforced by the computational technique?

Regarding biological relevance, I have one more comment: it has been demonstrated in Ref. 8 that particle stiffness is a very important determinant of uptake (stiff particles are taken up more easily than soft ones). Can the authors confirm this by using different stiffnesses? That would increase the biological relevance of their work.

Very minor comment regarding citations: Refs. 17 and 27 seem to be identical. The list of references for normal forces is not very exhaustive. For example, traction forces have also been measured on non-planar substrates. For planar ones, I suggest to include Kronenberg, Nils M., et al. "Long-term imaging of cellular forces with high precision by elastic resonator interference stress microscopy." *Nature cell biology* 19.7 (2017): 864. Also Ref. 29 could already be mentioned in this context in the introduction. For completeness, from the Campas group also Serwane, Friedhelm, et al. "In vivo quantification of spatially varying mechanical properties in developing tissues." *Nature methods* 14.2 (2017): 181 should be mentioned.

The supplementary videos seem to be missing and should be provided if the manuscript was resubmitted. It is one of the strength of this method that it does not require a reference configuration and thus easily can be used for live cell imaging.

Reviewer #3 (Remarks to the Author):

Review of "Superresolved microparticle traction force microscopy reveals subcellular force patterns in immune cell-target interactions" by Vorselen et al.

This paper describes the development of a deformable microparticle platform to analyze forces that expands current traction force microscopy capabilities to measure both shear and normal

stresses. A major advantage is that the particles can be seeded in a volume permitting strategies in traction force microscopy to be implemented locally by shape deformation analysis of such particles through conventional means such as confocal microscopy. The batch microparticle synthesis produces particles that are homogenous both in composition and between particles. The particles can be functionalized with specific ligands to elicit certain cell behavior, thus allowing specific cell interactions to be studied. They also feature attractive index of refraction properties to mitigate image distortions. The authors demonstrate the homogeneity and tunability of the microparticles, and computational calculations to correlate the particle shape with force sensing. The particles are thoroughly characterized in size, composition, functionalizability, and mechanical properties. The technique is applied to two well known cellular events to prove efficacy. The MPs are good local reporters of cell rigidity and deformation. The method can be implemented on conventional confocal microscope platforms and thus is appealing to a broad audience interested in cell mechanics. Overall, this appears to be a promising force sensing method developed by a strong laboratory and collaborators.

One of the most important aspects of this paper is that it is detailed enough for one to recreate the synthesis and characterization of the particles. The authors did an excellent job providing synthesis protocols, characterization, and providing a range of size and mechanical properties. They have very good control over size and refractive index. They also do a good job characterizing the mechanics, which is imperative for this application (figure 1k). It would be valuable to have more details on the spread in the young's modulus at the different crosslink ratios, as this directly relates to the stresses and forces determined from shape deformation of the MP's.

One of the problems with the method is that the force analysis based on shape deformation is very sophisticated. While it is mathematically detailed, it is not going to be straightforward for any lab without solid mechanics background to execute a similar analysis. Is it possible to provide some empirical relations or reference tables relating shape deformation (stretch of a certain percentage) to the expected force or stress on the particle? This may only be proper under ideal conditions and would need to be appropriately qualified, but a simplified interpretation of the shape deformations from ones confocal images without coding spherical harmonics etc would broaden the appeal of the method and make it applicable to more readers. At present it is too sophisticated and does not provide a pathway to make this accessible to the readership.

Specific Questions/Concerns:

- Were the two cellular events studied multiple times or just the one time that is displayed? Did the particles all show consistency in deformation/forces?
- First paragraph of the intro: "However, classical TFM is largely limited to in vitro applications..." but these particles are also limited in this way. The authors claim these particles are applicable in vivo, but it doesn't appear that is possible.
- Can the strategy be implemented "live" Is this effectively a dynamic tool or does it require fixing of the sample as in figure 3.
- Relating to the point above, there is a detail on page 7 that localization in z and x/y are determined to 40nm and 20nm. at "high signal to noise" In general there is a balance between resolution, field of view and speed. The paper should include more details surrounding how the image was acquired, was it a single image? What was the observation window? Formally resolution has units of position/sqrt Hz. We don't need resolution in these units but a reader will want to know what typical settings are required (observation window length) to obtain a similarly resolved image.
- On page 23 the details on "calculation of particle and surface properties" were confusing. What program was used to make the triangular mesh? Matlab? Python? Define all terms (including units) in equations such as Sphericity. More details on the curvature calculation would be helpful.

Minor Revisions:

- Last paragraph of the intro: "Finally, we solve the inverse problem of inferring the displacement

field and traction forces from the measured particle shape and traction-free regions." This was lengthy and confusing description. It should be combined with the subsequent sentence in a more active, direct way.

- Third paragraph in results: "Coating MPs with ligands is critical for triggering specific cellular behavior and conjugation with fluorescent molecules is required for visualization of the particles and their deformation in microscopy applications" could easily be shortened or combined with subsequent sentence.
- Second paragraph in intro: "However, current technologies have lacked the resolution to identify individual subcellular force transmitting structures" - but this particle system does not identify the structures either, it only shows the subcellular effect. Very minor but could be clarified.
- It would be good to know the observation window for figure 2.
- Text on page 8/9. The description is largely from a mechanics perspective. The readership will be interested in cellular forces. Based on the "Pa" and contact area it would be good to estimate forces and provide an example or two so that the readership can do this properly. For example, details surround a compressive stress of 50Pa over 1 μ m were provided. This could be converted to a ballpark force. Is there an example of tension as well? If it is not proper to do this then that would be good to note as well.
- Similar to the point above, on page 11/12 there is a 200Pa stress, perhaps this can be converted to a force if the indentation area is known.
- In figure 3 (Composite and Brightfield), the macrophage like cells (J774) are only partially visible. If the field of view permits, it would be good to somewhere show an image of the whole cell with the particle (perhaps in the supplemental).
- On page 16 the concentration of pMHC added is stated, but it would be beneficial to know the density or mention that it is saturated.
- Regarding deformations seen in multiple axis: Since a ring like compressive force will provide an apparent extension in an orthogonal axis of a deformable sphere, it would be good to mention this phenomena or caution the reader against interpreting that there is tension in this axis.

All in all, this paper offers a comprehensive analysis of this MP-TFM system and demonstrates efficacy. This system has great potential for future cell-cell force studies in a broad range of cell environments. The computational reconstruction of particles provides good visual of the forces present in different locations, and the algorithm can identify the magnitude of said forces. The computational methods in present form are a bit inaccessible for the broad readership, but this can be improved and clarified. The vast majority of the comments offered are minor and easily fixable. This is a strong force sensing methodology and, provided some adjustments are made, is of the quality needed for Nature Communications.

Reviewer 1 comments:

In this manuscript, Vorselen et al. describe the fabrication and implementation of a new microparticle-based traction force microscopy approach to measure forces during phagocytic engulfment and immunological synapse formation. The manuscript is well-written, and the data are interesting and high quality. In general, I believe the manuscript to be well suited for publication in Nature Communications; however, I have several concerns regarding the force reconstruction methods that would need to be addressed before I could recommend publication. These, as well as minor comments, are outlined below:

Major Comments:

1) The authors present a new force reconstruction approach to obtain both normal and shear forces from measurements of the deformed particle shape. As the authors note, because they only measure deformations at the surface and in the radial direction the problem of relating these deformations to surface tractions is underdetermined. To overcome this, they present an iterative optimization scheme that weights the residual radial displacement, the integrated traction on the region of the sphere thought to be traction-free, and the total elastic energy. The thought being, that the most correct solution will be lowest energy solution that matches the surface geometry yet doesn't predict substantial tractions on the traction-free region. I have several comments on this approach:

a. As this approach is substantially different from prior methods (which only report normal forces as in Campas et al. 2014 or measure the full displacement field as in Mohagheghian et al. 2018), the authors need to more fully demonstrate that their approach is capable of accurately reconstructing both shear and normal surface tractions and define where the limits of resolution lie. This could be done by simulating the forward problem under a variety of loads (both shear and normal) at various characteristic lengths, convolving the ideal surface with the point spread function of their instrument and overlaying Poisson noise and pixelation to simulate the imaging process, and then reconstructing using their standard protocol. This general approach has been outlined in prior studies (e.g. Sabass et al. Biophys J 2008, Legant et al PNAS 2012, Mohagheghian et al. Nature Communications 2018, Colin-York et al. Nanoletters 2016).

We followed the reviewer's suggestion and constructed various test cases. Our original computational strategy was unable to accurately recover traction forces in these test cases, which was likely due to the failure of finding the global minimum of our cost function due to the presence of local minima. To overcome this issue, we made major changes to our computational methodology, which are described in detail in the revised version of the main text, methods section, and the supplementary material. In brief, our revised computational method forces the calculated displacement field to provide an exact match to the experimentally measured shape, such that we now optimize only the elastic energy and residual tractions on the traction-free boundary, and a shape-difference term is no longer needed. This new methodology is faster, and results in more accurate fitting of the particle shape, with lower tractions on the traction-free region.

We tested the performance of our new methodology with test cases containing both normal and shear stresses, with loads of various length scales, and after addition of noise designed to mimic

the experimental noise. The results are presented in the new supplementary figure 9, and are discussed in the main text where we introduce our computational strategy.

b. Does the author's approach require the presence of a traction-free interface? E.G. could it still compute tractions for a fully engulfed microparticle?

We have performed our novel test cases with and without inclusion of a stress-free boundary and present the results in the new supplementary figure 9. It appears that the overall force pattern can still be recovered, even without using the knowledge of a stress-free region. However, this does lead to significant tractions on the traction-free region, and also larger error within the contact region. In particular, this seems to lead to large errors in normal forces when the scale of the deformations is large. We now caution the reader in the main text where we present the results of the fully internalized particle (particle 4) and in the figure caption of figure 5. We discuss these results further with the new supplementary figure 9.

c. Is the current approach limited to small-strains or can it be adapted to large strain/non-linearities? Some discussion on the limitations and applicable conditions would be helpful.

Our approach is currently limited to small-strains because the equilibrium condition (equation 1) that we used is based on Hooke's law and therefore only valid for small strains. Our approach is further limited to linear elasticity problems, since decomposition in spherical harmonics (as well as other forms of spectral analysis) are only valid for linear elasticity problems. We have clarified this in the methods section:

“For small-strain linear elasticity problems, the equilibrium condition (based on generalized Hooke's law) of the elastic continuum Ω in terms of the displacement field \mathbf{u} is:...”

and have clarified this in the main text. In addition, we added an explanation to the discussion: “Like most inverse TFM methods³⁷, our computational approach is only appropriate for linear elastic materials, and is currently also limited to small strains ($\epsilon \lesssim 0.1$), although it could be, in principle, extended for large strains.”

d. The 10Pa sensitivity claim in the abstract and discussion seems misleading to me. If I understand correctly, the 10 Pa value is derived from the residual tractions predicted on “traction-free” portion of the microparticle. However, as the authors note in their methods, this surface has been pre-smoothed by a user defined value to “prevent overestimation of high-frequency contributions” and further suppressed by the choice of alpha in the regularization scheme. It seems that this value could be made arbitrarily large or small by the choice of smoothing and regularization parameters. Moreover, the traction sensitivity in loaded regions will depend on the characteristic length of the loading and not simply on the traction amplitude (e.g. two opposing forces in close proximity would produce minimal surface displacement when averaged over a given resolution element and thus may not be detected regardless of their amplitude). In short, force sensitivity and traction length scale are inherently linked in TFM.

We agree with the reviewer that this statement was misleading. Due to the complexity of determining sensitivity in a general sense, we have removed such general claims throughout the manuscript.

e. Several other metrics are commonly used to assess solution accuracy including deviation from static (and rotational) equilibrium, but these aren't discussed here. Are these conditions enforced a-priori by the solution process or will the solutions deviate from these conditions (if so, how much)? In particular, the shear force calculations (e.g. Fig 5 c, second row, Fig 6) seem to show vortices of shear force that, if accurate, would exert a net rotational moment and cause the microparticle to rapidly spin. Intuitively, this does not seem correct. I also have trouble imagining how such patterns could be generated by the cell.

Translational and rotational equilibrium are enforced a priori, which we now state more clearly in the methods section text:

“This equation corresponds to the condition of static equilibrium in the absence of body forces, which are likely negligible compared to total surface traction forces.”

Hence, there is no spin of the particles. We had previously made a mistake in the plotting (not the calculation) of the shear forces, which resulted in the appearance of a net rotational moment. We have corrected all figures showing shear forces.

Minor Comments:

1) The super resolution claim in the title is a bit of a stretch. The authors use a similar Gaussian fitting procedure used in Campas et al. (compare Sup Fig 6 from this manuscript to Sup Fig 2 from Campas) to fit the microparticle surface with a precision below the conventional diffraction limit. More generally, the majority of TFM methods fit bead images by fitting a 2D or 3D Gaussian function to determine the bead centroid to well below the diffraction limit and thus “super-resolve” the displacement field. In this sense, nearly all TFM papers could be called “super-resolved” TFM.

We largely agree with the reviewer and have adapted the title and our claims at various places in the text. However, the starting point for our analysis is quite different than in the work by Campas et al., since we have a uniformly fluorescent particle, instead of a fluorescent marker of the edge of the droplet. It was, as far as we know, not obvious that this uniform particle stain would allow for such accurate shape determination. Moreover, fitting a Gaussian does not necessarily imply high edge localization precision, and this is especially true along the z-direction. (In the work by Campas et al, the apices of the droplets were omitted from analysis altogether.) Recovering the entire particle shape, including these apices, at high resolution is absolutely critical to analyzing forces for elastic microparticles. To our knowledge, we are the first to experimentally show that we can measure the entire particle surface with precision well below the diffraction limit (< 50 nm).

2) The introduction states “deformable microparticles have so far only been used as passive force reporters without biologically specific ligands”. Both Campas et al and Mohagheghian et al used microparticles functionalized with RGD. Campas also functionalized particles with cadherins. I would consider these to be biologically specific ligands.

We thank the reviewer for pointing this out and have removed this statement.

3) To assess particle homogeneity and mechanical uniformity, the authors imaged the radial distributions of conjugated fluorophores. A more rigorous control for isotropic/homogeneous mechanical properties would be to osmotically swell/shrink the particles and analyze the deviation from the starting geometry.

We performed the experiment suggested by the reviewer, and included the data in the new supplementary figure 3. As expected, with increased osmotic pressure, the particles appear smaller and still spherical. We also note that the soft ($C_c = 0.32\%$) particles for which we did the radial distribution imaging (Supplementary Fig. 2), are also significantly swollen (~5-fold in volume) from the initial emulsion droplet size. We have clarified this in the main text.

4) It would be amazing to see the traction dynamics during a live-cell movie either of phagocytosis or synapse formation. Fig 6 shows only three frames, all of which are already after the synapse is formed. Is there some technical limitation preventing this? Perhaps due to the computational resources required for traction calculation at many different time points? If so, this should be addressed in the discussion.

The primary challenge associated with obtaining and analyzing long time-scale (or high framerate) movies is related to the data acquisition. Firstly, since our field of view is limited in these high magnification experiments, it is challenging to capture events from the very start, and it is easier to find cells already interacting with a particle. Moreover, rapid 3D imaging is required and photobleaching, phototoxicity, and imaging artifacts present in advanced imaging techniques such as lattice light sheet microscopy, make it a challenge to conduct such experiments and collect images of the required quality at many timepoints. However, with some additional optimization of experimental conditions, or with some additional image processing to deal with artifacts, we believe that we will be able to capture data of the required quality to make such movies in the near future.

Currently the force calculations do require a significant amount of time for each shape (~ 2 hours). However, in movies, where subsequent frames will be similar, calculations are expected to be much faster since we can use the outcome of a previous frame as initial guess of the displacement field for subsequent frames. Hence, we believe that neither of these present fundamental limitations of the methodology.

We agree that it would be amazing to see the whole process from beginning to end and this will be a focus of our future efforts.

Reviewer 2 comments:

The authors report on a new method to measure cell forces through the deformation of soft hydrogel particles. The wider context of this study is traction force microscopy, which traditionally is performed on planar soft elastic substrates. Using beads rather than thick substrates is challenging but recently has been tackled by several groups. Motivated by their earlier Listeria assay for soft particles, the Otger Campas group has injected oil droplets into tissue and monitored its deformations (Refs. 13 and 14). Very recently, the Ning Wang group has used alginate hydrogel beads made in a microfluidic device to measure forces in tumor colonies (Ref. 15). A similar approach is described in a BioRxiv-paper by the Jochen Guck group (Ref. 36), which injected large (17 μm diameter polyacrylamide beads) into zebrafish.

The present work is similar its aims, but has several new elements that make it very appealing to the general audience interested in cell forces. (1) The authors do not use microfluidics (like in Refs. 15 and 36) to generate the particles, but extrusion through porous glass. This leads to relatively monodisperse particles in a size range from 4 and 15 μm . For many groups this approach might be easier to use than microfluidics. The rigidity range is similar as in the other studies, around kPa. (2) Great care is taken to determine particle shape very accurately, with sub-pixel (“super”) resolution. This also includes careful measurement of the refractive index. (3) A new computational method has been developed to infer traction stresses. In contrast to Refs. 15 and 36, the authors do not reconstruct the displacement field (either by bead tracking like in Ref. 15 or by a simple geometrical construction like in Ref. 36), but work only with the shape reconstructed in (2).

Because shape is not a unique predictor for the tractions, they define a cost function that is minimized in an iterative procedure. Here they use spherical harmonics as published in Ref. 17 by a subset of the present authors. This procedure seems to be impressive and rather rigorous, and sets this work clearly apart from the preprint Ref. 36. (4) Finally the method is applied to two situations of large biological importance, phagocytosis and the immunological synapse. In both cases, deformation and traction maps are presented and described.

Overall, this study is very solid and impressive, and a clear advance in pushing traction force microscopy into new territories. Its main strength is on the methods side, but there are also very interesting biological results contained in the two examples. In my view, this work is of large interest to the interdisciplinary and general audience of Nat Comms. I do have a few comments on how to further improve the quality and impact of this work.

We thank the reviewer for their kind words about our work.

The Young’s modulus is measured by AFM-experiments evaluated with the Hertz-model. I have two comments. First an independent measurement would be good. In Ref. 36, the authors compressed the beads by soluble dextran pressure and showed that the resulting bulk modulus together with the value of the Poisson ratio corresponds to the Young’s modulus. I suggest to repeat this procedure here. Second I wonder if there is a possible role of surface tension in this system. Obviously there is a surface tension in the oil phase which gives the spherical shapes, but I am more concerned with the increase in modulus after BSA-functionalization. Does the BSA

enter the beads ? If it stays at the interface, why should it change the modulus ? Should one not extend the Hertzian analysis by a surface tension term?

We have performed the suggested experiments by the reviewer and included a new supplemental figure (Supplemental Figure 6) based on the bulk modulus measurements. Together with the Young's moduli measured by AFM we were also able to make estimates of the Poisson's ratio of our particles (0.42 – 0.44), which is consistent with the measurements in ref. 36 (0.443) and indicates that the particles are almost incompressible.

Regarding the localization of BSA and the potential of a role of surface tension: We have shown in figure 1e and 1f that BSA localizes uniformly throughout the particle and hence BSA does not stay at the interface. This is also expected since the hydrodynamic radius of BSA (~ 3 nm) is much smaller than the pore size of our hydrogel nanoparticles (likely 10-50 nm). It is likely that BSA affects the hydrogel Young's modulus by creating additional crosslinks within the hydrogel, since BSA has multiple primary amines that could crosslink the carboxyl groups present in the gel and would result in an increase in Young's modulus. We have included an additional sentence in the text to clarify that the increased Young's modulus is likely due to additional crosslinking of the hydrogel.

The biological results are very interesting, but somehow phenomenological. Why do the authors not present the results of actin and/or myosin inhibitions to prove which machinery is behind these forces (like Latrunculin A in Ref. 15) ? I also wonder why actin imaging has only been used for the immunological synapse and not for the phagocytosis ? Finally I think that the shear force patterns presented in Figs. 5 and 6 should be interpreted. Why do they both show vertex patterns ? It is clear from the hairy ball theorem that some singularities have to exist, but why should there be chirality in the problem ? Is this something enforced by the computational technique ?

We agree with the reviewer that experiments with pharmaceutical/genetic perturbations will be extremely interesting. However, we believe that dissecting the underlying molecular mechanisms giving rise to the force profiles requires a complete study by itself. In our opinion, such a study would necessarily encompass higher throughput analysis of cellular induced forces, analysis of particle-to-particle variation, correlative studies of protein localization with particle deformation/cellular forces, and pharmacological/genetic perturbations. We believe this to be beyond the scope of the current manuscript, which largely focuses on establishing the technology. The reason for the lack of actin images for the phagocytosis work is that we didn't perform this staining and imaging for these samples. We agree that actin images would be an interesting addition, however, we believe that the existing data are interesting as well.

Following the advice from the reviewer, we expanded our discussion the shear force patterns. Previously, we had made a mistake in the plotting of the shear forces, which gave rise to the vertex patterns. We apologize for this mistake, and have corrected all figures showing shear forces.

Regarding biological relevance, I have one more comment: it has been demonstrated in Ref. 8 that particle stiffness is a very important determinant of uptake (stiff particles are taken up more easily than soft ones). Can the authors confirm this by using different stiffnesses ? That would increase the biological relevance of their work.

We performed the experiment suggested by the reviewer, and, like previous studies, we find that phagocytic uptake is strongly affected by target rigidity and more efficient for stiffer targets. We now show this result in supplementary figure 8 and mention our findings in the main text.

Very minor comment regarding citations: Refs. 17 and 27 seem to be identical. The list of references for normal forces is not very exhaustive. For example, traction forces have also been measured on non-planar substrates. For planar ones, I suggest to include Kronenberg, Nils M., et al. "Long-term imaging of cellular forces with high precision by elastic resonator interference stress microscopy." *Nature cell biology* 19.7 (2017): 864. Also Ref. 29 could already be mentioned in this context in the introduction. For completeness, from the Campas group also Serwane, Friedhelm, et al. "In vivo quantification of spatially varying mechanical properties in developing tissues." *Nature methods* 14.2 (2017): 181 should be mentioned.

We thank the reviewer for pointing out the duplicate reference, and these papers. We have removed the duplicate reference and included the suggested references.

The supplementary videos seem to be missing and should be provided if the manuscript was resubmitted. It is one of the strength of this method that it does not require a reference configuration and thus easily can be used for live cell imaging.

We apologize for the lacking supplementary movies. They are included with this resubmission.

Reviewer 3 comments:

This paper describes the development of a deformable microparticle platform to analyze forces that expands current traction force microscopy capabilities to measure both shear and normal stresses. A major advantage is that the particles can be seeded in a volume permitting strategies in traction force microscopy to be implemented locally by shape deformation analysis of such particles through conventional means such as confocal microscopy. The batch microparticle synthesis produces particles that are homogenous both in composition and between particles. The particles can be functionalized with specific ligands to elicit certain cell behavior, thus allowing specific cell interactions to be studied. They also feature attractive index of refraction properties to mitigate image distortions. The authors demonstrate the homogeneity and tunability of the microparticles, and computational calculations to correlate the particle shape with force sensing. The particles are thoroughly characterized in size, composition, functionalizability, and mechanical properties. The technique is applied to two well known cellular events to prove efficacy. The MPs are good local reporters of cell rigidity and deformation. The method can be implemented on conventional confocal microscope platforms and thus is appealing to a broad audience interested in cell mechanics. Overall, this appears to be a promising force sensing method developed by a strong laboratory and collaborators.

One of the most important aspects of this paper is that it is detailed enough for one to recreate the synthesis and characterization of the particles. The authors did an excellent job providing synthesis protocols, characterization, and providing a range of size and mechanical properties. They have very good control over size and refractive index. They also do a good job characterizing the mechanics, which is imperative for this application (figure 1k). It would be valuable to have more details on the spread in the young's modulus at the different crosslink ratios, as this directly relates to the stresses and forces determined from shape deformation of the MP's.

One of the problems with the method is that the force analysis based on shape deformation is very sophisticated. While it is mathematically detailed, it is not going to be straightforward for any lab without solid mechanics background to execute a similar analysis. Is it possible to provide some empirical relations or reference tables relating shape deformation (stretch of a certain percentage) to the expected force or stress on the particle? This may only be proper under ideal conditions and would need to be appropriately qualified, but a simplified interpretation of the shape deformations from ones confocal images without coding spherical harmonics etc would broaden the appeal of the method and make it applicable to more readers. At present it is too sophisticated and does not provide a pathway to make this accessible to the readership.

We agree with the reviewer that the derivation of cellular forces is a complex problem. It is not obvious to us that a more straight-forward approach allowing simplified interpretation of the forces is possible, and it surely would not be an easy task to develop such an approach. However, we have tried to make our approach as accessible as possible by making all our code publicly available. Hence, other researchers will not have to code spherical harmonics themselves to derive forces, but will be able to implement our code to derive shapes and forces from confocal image stacks.

Specific Questions/Concerns:

- Were the two cellular events studied multiple times or just the one time that is displayed? Did the particles all show consistency in deformation/forces?

We have included the following statement: “Images are representative examples from multiple ($n > 5$) independent experiments.” We have consistently seen similar behavior in both fixed and live cell experiments. However, we believe that the in depth analysis of these larger data sets is beyond the scope of the current manuscript and we plan to make this the focus of future work.

- First paragraph of the intro: “However, classical TFM is largely limited to in vitro applications...” but these particles are also limited in this way. The authors claim these particles are applicable in vivo, but it doesn't appear that is possible.

We didn't show in vivo application of our methodology in this manuscript. However, as shown by for example Träber et al (bioRxiv 420844v1), polyacrylamide particles are, in fact, amendable to in vivo approaches by injection of the particles into living organisms such as zebrafish. We have addressed this in the second paragraph of our introduction, as well as in the last paragraph of our discussion.

- Can the strategy be implemented “live” Is this effectively a dynamic tool or does it require fixing of the sample as in figure 3.

In figure 6 we show live implementation of the methodology, where we have performed multiple force measurements with a living T cell. We now mention in the subheading, figure caption, intro, results and discussion that these were live cell experiments.

- Relating to the point above, there is a detail on page 7 that localization in z and x/y are determined to 40nm and 20nm. at “high signal to noise” In general there is a balance between resolution, field of view and speed. The paper should include more details surrounding how the image was acquired, was it a single image? What was the observation window? Formally resolution has units of position/sqrt Hz. We don't need resolution in these units but a reader will want to know what typical settings are required (observation window length) to obtain a similarly resolved image.

We have included the requested experimental details in the methods section: “Edge localization precision and reference shape measurements were performed with a 50 mW 560 nm laser, where laser power output was varied between 1 – 50% and image stacks were acquired with 50 ms exposure per slice ($144 \mu\text{m} \times 144 \mu\text{m}$ each).”

- On page 23 the details on “calculation of particle and surface properties” were confusing. What program was used to make the triangular mesh? Matlab? Python? Define all terms (including

units) in equations such as Sphericity. More details on the curvature calculation would be helpful.

We revised this paragraph. We now state that these calculations were performed in Matlab, and we now define all terms. Because sphericity is unitless, any consistent units for length (e.g. pixels, or μm) are appropriate for calculating sphericity. For the curvature calculations, we perform calculations that were described previously and we provide references to this earlier work. Moreover, we provide the Matlab code used for the curvature calculations (along with all other Matlab code used for the data analysis in the manuscript) and provide a direct link to this code under the section "Code Availability".

Minor Revisions:

- Last paragraph of the intro: "Finally, we solve the inverse problem of inferring the displacement field and traction forces from the measured particle shape and traction-free regions." This was lengthy and confusing description. It should be combined with the subsequent sentence in a more active, direct way.

We have rephrased the sentence.

- Third paragraph in results: "Coating MPs with ligands is critical for triggering specific cellular behavior and conjugation with fluorescent molecules is required for visualization of the particles and their deformation in microscopy applications" could easily be shortened or combined with subsequent sentence.

We have shortened this sentence.

- Second paragraph in intro: "However, current technologies have lacked the resolution to identify individual subcellular force transmitting structures" - but this particle system does not identify the structures either, it only shows the subcellular effect. Very minor but could be clarified.

We adapted this sentence to: "However, current technologies have lacked the resolution to identify contributions from individual subcellular force transmitting structures".

- It would be good to know the observation window for figure 2.

We now have included the exposure time in the figure captions, as well as the detailed acquisition settings (including FOV and exposure time) in the methods section.

- Text on page 8/9. The description is largely from a mechanics perspective. The readership will be interested in cellular forces. Based on the "Pa" and contact area it would be good to estimate

forces and provide an example or two so that the readership can do this properly. For example, details surround a compressive stress of 50Pa over 1 μ m were provided. This could be converted to a ballpark force. Is there an example of tension as well? If it is not proper to do this then that would be good to note as well.

- Similar to the point above, on page 11/12 there is a 200Pa stress, perhaps this can be converted to a force if the indentation area is known.

We now include regional force estimation by integration over the particle surface in various places in the result section.

- In figure 3 (Composite and Brightfield), the macrophage like cells (J774) are only partially visible. If the field of view permits, it would be good to somewhere show an image of the whole cell with the particle (perhaps in the supplemental).

We followed the reviewer's suggestion and included images in which the entire cell is visible in Supplementary Figure 8. We refer to this figure at the appropriate place in the main text, as well as in the figure caption of figure 3.

- On page 16 the concentration of pMHC added is stated, but it would be beneficial to know the density or mention that it is saturated.

Saturation of the binding groups in the particle is unlikely to be reached, because of the porosity of the bead and the high concentration of carboxylated groups within them. Due to the porous nature of the bead it is also non-trivial to directly measure the surface density that the cell is exposed to. However, to address this point we have performed additional experiments that show that the concentration of pMHC that is added saturates the T-cell response as evaluated by CD69 expression. These new data are presented in Supplementary figure 11.

- Regarding deformations seen in multiple axis: Since a ring like compressive force will provide an apparent extension in an orthogonal axis of a deformable sphere, it would be good to mention this phenomena or caution the reader against interpreting that there is tension in this axis.

This is indeed a very important point, and have clarified it in the results section as well as with the discussion of our novel test cases in the supplementary material which illustrate this effect (Supplementary Fig. 9).

All in all, this paper offers a comprehensive analysis of this MP-TFM system and demonstrates efficacy. This system has great potential for future cell-cell force studies in a broad range of cell environments. The computational reconstruction of particles provides good visual of the forces present in different locations, and the algorithm can identify the magnitude of said forces. The computational methods in present form are a bit inaccessible for the broad readership, but this can be improved and clarified. The vast majority of the comments offered are minor and easily

fixable. This is a strong force sensing methodology and, provided some adjustments are made, is of the quality needed for Nature Communications.

REVIEWERS' COMMENTS:

Reviewer #1 (Remarks to the Author):

The authors have addressed all of my prior comments. I believe that the revised manuscript is suitable for publication in Nature Communications and will be a solid contribution to the field. In particular, the methods for high-throughput microparticle generation will hopefully enable other groups to adopt these measurements in their own labs.

One minor suggestion would be to include a few paragraphs in a supplementary note presenting a balanced discussion of the advantages and disadvantages of the various methods for reconstructing tractions on the surface of microparticles (e.g. green's function, finite element optimization, adjoint methods etc). I would by no means expect this to be a full review of the literature, but some additional discussion would be helpful to place this new method in the field. Presumably, it was the tradeoffs of these other methods that motivated the development of the approach presented here, and it would be helpful for the reader and traction force community to better understand these motivations.

Additionally, it would be helpful to add some discussion of how the current approach could be improved in the future (perhaps combining surface measurement with tracer tracking within the particles for even higher-resolution, extension to non-linear materials or large displacements, or computational parallelization/GPU implementation for higher throughput/speed)?

-Wesley Legant

Reviewer #2 (Remarks to the Author):

The authors have responded well to the comments of the reviewers. In particular, they have provided TFM test cases, improved their computational methodology, identified the vortex patterns as artefacts, used particles of different stiffnesses and obtained biological meaningful results (stiff particles are taken up faster), estimated the Poisson ratio and provided the movies (which are great). As far as I can see, all concerns were taken care of in a constructive manner and several weaknesses have been removed. The authors also clearly and convincingly state which suggestions are left for future work. I now recommend acceptance.

Reviewer #3 (Remarks to the Author):

This paper presents a very interesting and powerful new method to probe cellular forces with deformable particles that has many applications and will reveal new biology. The authors have addressed all concerns raised earlier. They have documented their approach well for both a reader new to cellular forces and an expert. It is a very exciting paper.

I believe the manuscript should be accepted. I look forward to seeing it published and introducing this new method to my students!

Sincerely,
Matt Lang

Reviewer 1 comments:

The authors have addressed all of my prior comments. I believe that the revised manuscript is suitable for publication in Nature Communications and will be a solid contribution to the field. In particular, the methods for high-throughput microparticle generation will hopefully enable other groups to adopt these measurements in their own labs.

One minor suggestion would be to include a few paragraphs in a supplementary note presenting a balanced discussion of the advantages and disadvantages of the various methods for reconstructing tractions on the surface of microparticles (e.g. green's function, finite element optimization, adjoint methods etc). I would by no means expect this to be a full review of the literature, but some additional discussion would be helpful to place this new method in the field. Presumably, it was the tradeoffs of these other methods that motivated the development of the approach presented here, and it would be helpful for the reader and traction force community to better understand these motivations.

Additionally, it would be helpful to add some discussion of how the current approach could be improved in the future (perhaps combining surface measurement with tracer tracking within the particles for even higher-resolution, extension to non-linear materials or large displacements, or computational parallelization/GPU implementation for higher throughput/speed)?

A suggested by the reviewer, we have added a few paragraphs of Supplementary Discussion on the advantages and disadvantages of various methods for reconstructing tractions forces, as well as some ideas for how the technology presented in this paper can be improved further.

Reviewer 2 comments:

The authors have responded well to the comments of the reviewers. In particular, they have provided TFM test cases, improved their computational methodology, identified the vortex patterns as artefacts, used particles of different stiffnesses and obtained biological meaningful results (stiff particles are taken up faster), estimated the Poisson ratio and provided the movies (which are great). As far as I can see, all concerns were taken care of in a constructive manner and several weaknesses have been removed. The authors also clearly and convincingly state which suggestions are left for future work. I now recommend acceptance.

We thank the reviewer for their kind words about our work.

Reviewer 3 comments:

This paper presents a very interesting and powerful new method to probe cellular forces with deformable particles that has many applications and will reveal new biology. The authors have addressed all concerns raised earlier. They have documented their approach well for both a reader new to cellular forces and an expert. It is a very exciting paper.

I believe the manuscript should be accepted. I look forward to seeing it published and introducing this new method to my students!

We thank the reviewer for their very kind words about our manuscript.